# Revisiting System-Heterogeneous Federated Learning through Dynamic Model Search

## Abstract

Federated learning is a distributed learning paradigm in which multiple distributed clients train a global model while keeping data local. These clients can have various available memory and network bandwidth. However, to achieve the best global model performance, how we can utilize available memory and network bandwidth to the maximum remains an open challenge. In this paper, we propose to assign each client a subset of the global model, having different layers and channels on each layer. To realize that, we design a constrained model search process with early stop to improve efficiency of finding the models from such a very large space; and a data-free knowledge distillation mechanism to improve the global model performance when aggregating models of such different structures. For fair and reproducible comparisons between different solutions, we directly allocate different memory and bandwidth to each client according to memory and bandwidth logs collected on devices. The evaluation shows that our solution can have accuracy increase ranging from 2.43% to 15.81% and provide 5% to 40% more memory and bandwidth utilization with negligible extra running time, comparing to existing state-of-the-art system-heterogeneous federated learning methods under different available memory and bandwidth, non-i.i.d. datasets, image and text tasks.

## 1 Introduction

Federated learning (FL) (McMahan et al., 2017) is a distributed learning paradigm to train a global model over multiple distributed devices. With these devices' edge computation power, network communication, and a centralized server, the system keeps all users' data local, ensuring privacy. As cloud servers are powerful, the edge device's computation and network bandwidth become training bottlenecks. To address this, efforts focus on optimizing edge computation and communication through model compression (Li et al., 2021; Wu et al., 2020; Luo et al., 2021; Shen & Chen, 2020), adaptive batch sizes (Wang et al., 2019), scheduling (Yao et al., 2021b), regularization (Acar et al., 2020), and federated neural architecture search (Yao et al., 2021a; He et al., 2020; Garg et al., 2021; Laskaridis et al., 2022; Yao & Li, 2024), etc.

To address the issue that different devices can have different memory, bandwidth, etc., system-heterogeneous federated learning methods are proposed to assign models of different sizes or different computation complexities (e.g. Flops) to each device (Horvath et al., 2021; Diao et al., 2021; Alam et al., 2022; Caldas et al., 2018b; Shen & Chen, 2020; Li et al., 2023b). Previous solutions were based on the assumption that each device's running status (e.g. network connection, memory usage, etc.) would not change during the federated learning process. But in the real world, such an assumption will no longer hold (Yao et al., 2021b; Guo et al., 2021; Wang et al., 2020; Huang et al., 2020; Almeida et al., 2022).

However, in practical applications, there is no guarantee that every time a device checks in, its available memory or bandwidth will remain constant. For instance, during the training process, a mobile phone's memory resources may become constrained if a user leaves models training in the background, and later closes the application. Conversely, if a device consistently has low memory resources, it may never check in.

Another issue with existing research on system heterogeneity is the lack of direct model assignment to each client based on available memory during evaluation. Prior studies have often relied on simulation approaches

where a fixed number of devices are assigned designated model complexities in each communication round. However, such simulations may not accurately reflect performance in practical scenarios.

Different from previous evaluation processes, we develop a system-heterogeneous federated learning system. Our system can simulate the clients' available resources (memory, bandwidth, etc.) with logs collected on physical devices (e.g. network speed, memory logs). With our system, we can have reproducible evaluation over different algorithms fairly with the same configuration files. We argue that we should perform experiments directly over changing resources such as memory and bandwidth available on the devices, and that the selection of models assigned to clients is based on their real-time available memory and bandwidth.

On running our system, we find that existing solutions may achieve sub-optimal results in certain settings. We observe that existing solutions adopted the same channel pruning rate on each layer. The heterogeneity in models is only based on one dimension of different channel numbers. This leads to the consequence that we have a limited range of selecting different model structures for each client. Hence, each client may not receive the model fitting into its memory or bandwidth the best, resulting in sub-optimal performance.

Motivated by these empirical observations, we propose directly assigning models of different structures to devices based on their available memory and bandwidth in each round. These models vary in channel numbers per layer and the number of layers. With more structure choices and finer-grained model selection, our method can cover a broader range of cases across devices in federated learning. The large flexibility of model structures results in a vast search space. Therefore, we propose a constrained search method, gradually expanding the search space through random search.

After the model aggregation stage on the server, we further develop an original federated in-place distillation method to improve the performance. Different from conventional knowledge distillation (Hinton et al., 2015) methods, we do not need any extra data (neither generated nor public datasets) to conduct distillation. Instead, we conduct knowledge distillation between subsets of the global model and the global model itself on the server. The proposed distillation module can also be applied to previous system-heterogeneous methods.

With our system, we revisit various federated learning settings and train different models including convolution neural networks and transformers over i.i.d. and non-i.i.d. datasets. In the evaluation, we test the system with bandwidth logs of running HTTP/2 applications. It turns out that our models have a better utilization of bandwidth and memory over clients up to 40%. It is shown that we can improve the accuracy by 2.43% to 15.81% among image and text classification tasks.

## 2 Related Work and Motivation

### 2.1 System Heterogeneity in Federated Learning

In the context of federated learning (FL), we have $N$ clients on devices. Each client has its own local data $\mathcal{D}_i, i \in [N]$. The data on these clients can be either i.i.d. or non i.i.d. The server aims to train a global model **W** which can be utilized by all these clients.

In a practical FL system, devices often face resource constraints, and many cannot run a large global model. This disparity in resources is called *system heterogeneity*. Memory usage and bandwidth are common resource budgets (Yao et al., 2021b). As existing edge devices are equipped with GPUs (Jetson TX1, iPhone, MacBook, etc. ), we mainly consider GPU memory. We do not consider energy, as users are unlikely to frequently change energy modes during the learning process.

Resource budgets can fluctuate during federated learning as users run applications (Yao et al., 2021b; Guo et al., 2021; Wang et al., 2020; Huang et al., 2020; Almeida et al., 2022), raising the need for a system to handle it. As shown in Figure 1, we measure the available memory on the Apple MacBook Pro with M1 chip while running an image generation application ControlNet (Zhang & Agrawala, 2023). The available memory will fluctuate from 2GB to 10GB. We also consider the transmission rate, Many edge devices have the bandwidth (Diao et al., 2021; Alam et al., 2022; Caldas et al., 2018b; Li et al., 2021; Yao et al., 2021a) limitation and cannot transmit so many model parameters at the same time. The transmission rate or the network speed will change frequently (Yao et al., 2021b; Guo et al., 2021). In Figure 1, we visualize the

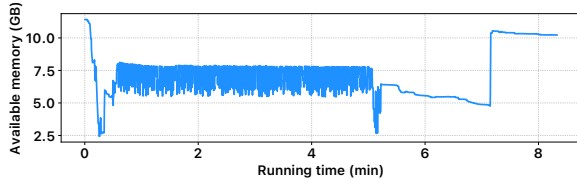 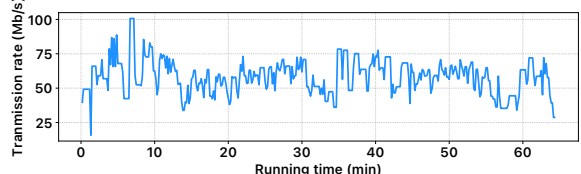

Figure 1: The changing of available memory and transmission rate when running the applications.

bandwidth logs recorded on the MacBook using a public Wi-Fi of 5GHz bandwidth. The network transmission rate can frequently change on user-end devices.

## 2.2 More Flexibility in Different Model Structures Can Provide Better Performance

Though each client device is not able to run the largest model which has the best accuracy, we still want to learn a global model having as high accuracy as possible. Conventional methods in efficient FL (Li et al., 2021; Wu et al., 2020; Luo et al., 2021; Yuan et al., 2022; Shen & Chen, 2020; Wang et al., 2019) did not base their optimization over actual resource constraints on each client, making the assigned models still possibly fail to run on the devices. Hence, a series of system-heterogeneous federated learning methods are proposed. HeteroFL (Diao et al., 2021) leveraged channel pruning to assign models of different channel widths. FjORD (Horvath et al., 2021) further proposed an ordered dropout knowledge distillation module. FedDropout and Split-Mix FL (Caldas et al., 2018b; Hong et al., 2022) leveraged similar idea but they selected the pruned channels randomly. Dun et al. (2023) expanded the idea of randomly selecting pruned channels through Dropout layers and applied this idea into asynchronous scenario. FedRolex (Alam et al., 2022) further introduced a rolling scheme to the pruning process, applying a more balanced node selection policy. AnyCostFL (Li et al., 2023b) introduced a mechanism that in each communication round, channels are first sorted on basis of their importance and each client choose top-$k$ important channels. However, all these previous methods choose to only prune the channel of each client model at the same rate for all layers.

Assigning clients with models of different model structures can bring much more flexibility. In a toy experiment with 100 clients, 10 selected per round, we used a non-i.i.d. CIFAR10 (Krizhevsky et al., 2009) dataset with a Dirichlet distribution ($\alpha = 0.1$) and a local epoch of 5. A ResNet152 model pruned by 62% channels per layer, achieving a similar size to ResNet18, only reached **68.73%** accuracy compared to ResNet18's **71.91%** and the original ResNet152's **74.51%**. Thus, using the smallest models on all clients or channel pruning is suboptimal; assigning models of different structures to each client is intuitively better.

Federated neural architecture search (NAS) (Yao & Li, 2024; Yao et al., 2021a; He et al., 2020; Yuan et al., 2022; Laskaridis et al., 2022) is a method for model search in federated learning, inspired by automating the design of artificial neural networks. Existing methods are either tailored for personalized federated learning (Yao & Li, 2024; Laskaridis et al., 2022) or require transmitting and computing large supernets on clients (He et al., 2020; Yuan et al., 2022; Yao et al., 2021a). A major difference between our work and NAS is that we do not aim to automate finding the most suitable model structures for each device or the global model. Instead, we focus on leveraging constrained memory and network bandwidth on devices to train a given global model in federated learning. Existing federated NAS methods fail to accommodate such use cases.

## 2.3 In-place Distillation

In centralized neural architecture search, employing in-place distillation enhances the supernet's performance significantly (Yu & Huang, 2019; Yu et al., 2020). It is used to improve the accuracy of the super-net where the super-net is aggregated from a lot of sampled sub-networks. In our federated learning setting, we have a similar case where we need to aggregate several sampled sub-networks into a global big network. Hence, we are motivated to bring the in-place distillation technique into federated learning. In one-stage NAS supernet training, multiple sub-networks are sampled at each training step. In the centralized NAS approach, the entire supernet is initially trained directly on the dataset using real labels. Subsequently, the sampled sub-networks are trained using logits from the supernet rather than real labels. The NAS then aggregates all sampled

networks into the supernet. In summary, denoting the supernet as $\mathbf{W}$, the sampled sub-networks as $\mathbf{W}_s$ (representing this set), and the centralized dataset as $\mathbb{D}_c$, the supernet is trained using a specific loss function $\mathcal{L}_{NAS}$:

$$\mathcal{L}_{NAS} = \mathcal{L}_{\mathbb{D}_c}(\mathbf{W}) + \gamma \mathbb{E}_{\mathbf{W}_s} \mathcal{L}_{\text{KD}}([\mathbf{W}, \mathbf{W}_s]; \mathbf{W}^{t-1}) \tag{1}$$

where $\mathbf{W}^{t-1}$ means the weights of the supernet in the last iteration and $\mathcal{L}_{\text{KD}}$ is the distillation loss, e.g. Kullback–Leibler divergence. However, we can not directly leverage such a process into the model search process in the federated learning setting. Because the data is safely kept and distributed over the clients, we can neither conduct the first step of training the supernet with a centralized dataset nor the second step of training subnets with soft labels. FjORD (Horvath et al., 2021) proposed a self-distillation process, conducted on the clients. However, as data is probably non-i.i.d. in FL, such an operation can misguide the distilled model overfit on local datasets and cause severe performance degradation. Besides, carrying out distillation process on the clients will add on more overhead on clients.

Other solutions for system-heterogeneous federated learning based on knowledge distillation were also proposed. FedMD (Li & Wang, 2019) can aggregate client models of different user self-designed structures. FedDF (Lin et al., 2020) aggregates the client models and the global model through knowledge distillation over unlabeled public data. Fed-ET (Cho et al., 2022) proposed a weight consensus distillation scheme with diversity regularization. (Zhu et al., 2021) proposed to train a data generator on the server to generate data for knowledge distillation. However, these methods rely on public or generated data. Furthermore, in these methods, client model weights are partially or entirely transmitted to the server. Such design are incompatible with secure aggregation protocols violating the initial design of FL.

## 3 Dynamic Model Pruning for System Heterogeneous FL

While intuitively, extending HeteroFL, FedRolex, etc., from simple channel pruning to searching models of different structures should improve performance, implementation is challenging. We show the overall workflow in Figure 2. The algorithm version is in Appendix A. The server holds the global model. In each communication round, the server will sample single models of different structures with **resource constrained model search** and then send them to each client participated in this round. Each client will conduct local training with assigned models and their own data for several local epochs, the same as the process in FedAvg. After that, they upload the trained model weights as well as the information about available memory and bandwidth to the server. The server will first aggregate models of different structures into the global model. We will conduct several iterations of **federated in-place distillation** on the server to improve the global model performance. The system will then run into the next round.

### 3.1 Global Model Design and Aggregation

We build the global model into several search dimensions including depths, widths, etc. For each sampled client model, a feature dimension will be searched. If it is 0, it means such a layer is not selected.

**Definition 3.1.** (The global model and search space) Given a global model $\mathbf{W}$ with $d$ layers. We have a search space $(\mathbf{W}, \mathbb{S})$ where $\mathbb{S} = \{s_1, \cdots, s_m\}$, a ratio set for the search options on each layer ($m \geq 1$). Each search option defines the output dimension of the layers . $\mathbb{S}$ has least one element $s_m = 1$ representing no change in the original operation.

The $\mathbb{S}$ in the search space can either be a continuous (e.g. channel pruning rate in $(0,1]$) or a discrete space.

**Definition 3.2.** (The sampled client model) With a given search space $(\mathbf{W}, \mathbb{S})$. We can represent a subnet $\mathbf{W}_i$ as $\mathbf{W} \otimes \mathbf{V}$ where $\mathbf{V} = (\boldsymbol{v}_1, \ldots, \boldsymbol{v}_d)$ and $d$ is the number of the layers of $\mathbf{W}$, except for the last layer, $\boldsymbol{v}_i \in \mathbb{S}, \forall i \in [d-1]$.

In Figure 3, we give three examples. The input channel of the first layer depends on the input and the output channel of the last layer depends on the output. The output channel of each layer is decided by $\boldsymbol{v}_i$ and the input channel of each layer is decided by $\boldsymbol{v}_{i-1}$. For the first example, it is the same as the model generated in previous methods where $\boldsymbol{v}_i$ are all the same. For the second example, each layer has a different prune rate. For the third example, we can change the depth of the model, and each layer is removable.

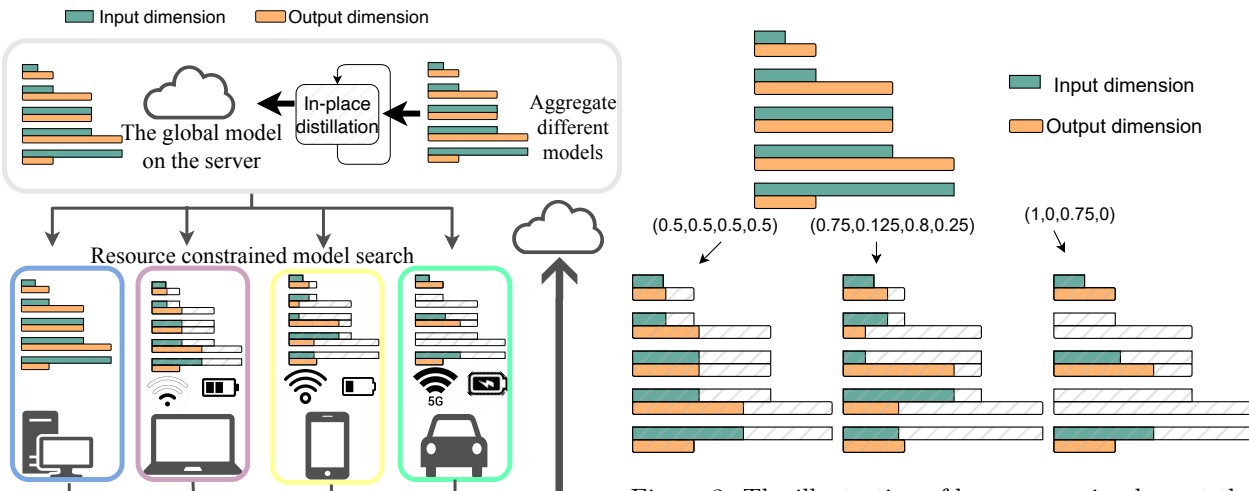

Figure 2: The overall workflow of our method to train the global model in system-heterogeneous FL.

Figure 3: The illustration of how we can implement the models assigned to the client according to definition 3.2.

As a result, we can use $(\mathbf{W}, (1, \ldots, 1))$ to represent the global model. The previous methods of only one dimension of channel pruning rate can be viewed a special case of our problem where the search space only has $m$ choices: from $(\mathbf{W}, (s_1, \ldots, s_1))$ to $(\mathbf{W}, (s_m, \cdots s_m))$.

### 3.1.1 Sample and Initiliaze a Client Model

We construct the client model based on our representation of $(\mathbf{W}, \mathbf{V}_i)$. With the given search space, we sample a subnet for each client in every communication round. The structure of the sampled subnet will be decided by $\mathbf{V}_i$. If the original hyper-parameter to decide $j$th layer $(j \in [d-1])$ in the global model is $l$, e. g. output channel width, the hyper-parameter of output channel width in the newly sampled sub-net is $v_j l$. In a sampled subnet, the weights of a layer are copied from the global model is $\mathbf{W}$. As each subnet $\mathbf{W}_i$ is a subset of the global model $\mathbf{W}$, for each parameter, we find the corresponding value in the global model and copy to initialize the subnet.

### 3.1.2 Aggregation Method

The next challenge is aggregating models of different structures into the global model after the client uploads them. In aggregation stage of conventional federated learning, we can add parameters at the same position together and divide the number of clients participated However, since the models have different structures, traditional aggregation solution does not work. In previous work (Yao et al., 2021a; Laskaridis et al., 2022), this is done at the operation level. For example, for a convolution layer, we have two choices of kernels sizes, one is $3 \times 3$ and the other is $5 \times 5$, like what is shown in Figure 4. Previous methods treat $3 \times 3$ and $5 \times 5$ convolution layers as separate operations and aggregate their parameters independently using the same add and divide method in conventional federated learning. In such a way, client 2 will only contribute to the $5 \times 5$ kernel but not $3 \times 3$ kernels.

To address such a problem, we perform averaging at the element-wise level. For instance, as illustrated in Figure 4, In our approach, the parameters of $3 \times 3$ convolution layers are aggregated together with those of $5 \times 5$ kernels. For each parameter in the global model, we check whether each client has such a parameter. Then, we add the parameters of all clients who have such a parameter together and divide by the number of clients who have such a parameter. For example, in Figure 4, the client 1 and 3 only have parameters in the center $3 \times 3$. So, for the parameters in the center, they are added over three clients and divided by 3. The parameters in the corner are only added over client 2 and divided by 1.

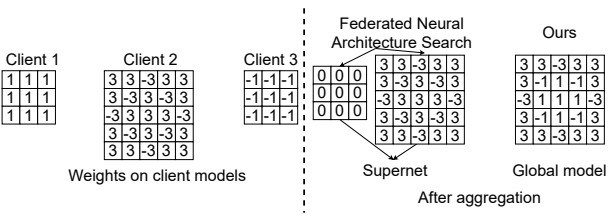
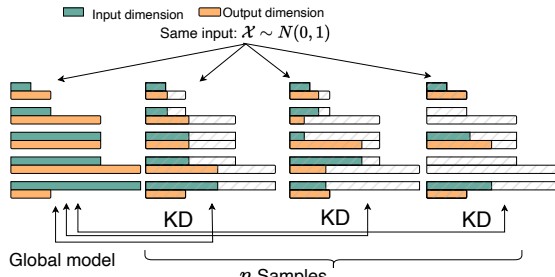

Figure 4: The illustration of our aggregation method of parameters from different clients. The example shows one convolution layer with different kernel sizes on the clients.

Figure 5: The process of in-place distillation on the server. Notably these $n$ new structures are not sampled from $\mathbb{P}$ but from $(\mathbf{W}, \mathbb{S})$.

### 3.1.3 Convergence Analysis

**Theorem 3.3.** *Model search-based and model-heterogeneous-based system-heterogeneous federated learning methods shares the same convergence rate as FedAvg in O notation of complexity, which is $O(\frac{1}{T})$.*

Compared to conventional federated learning, our expanded search space entails traversing more strcutures (from $m-1$ possibilities to $m^d$) and aggregating models of different structures, yet this does not adversely affect convergence. The proof can be found in Appendix C.

### 3.2 Resource Constrained Model Search

During the sampling phase, we select the largest model that satisfies each client's resource condition from the search space. By 'largest,' we mean the structure with the most parameters. Apart from the number of parameters, there are also other criteria such as GPU utilization and efficiency. This criteria is more like a balance criteria to best meet all requirements and achieve the best performance. Hence, we choose the criteria which achieves the best tradeoff, which is the number of parameters. We will later show this. However, due to the larger flexibility provided by the expansive search space, a challenge arises: it is impossible to enumerate all possible structures (a total of $m^d$ possible structures) and search for one that meets the requirements.

As a result, instead of exhaustively listing all possible structures, we select models from a sampling pool. We gradually expand the sampling pool through the training process. Initially, the sampling pool $\mathbb{P}$ consists of only the largest model (the global model) and the smallest sub-net. When sampling a client model at each step, we first traverse the sampling pool to find the largest structure that satisfies the requirements. To expand the sampling pool and increase the number of candidates, we adopt the searched structure with a probability of $\epsilon$, and with a probability of $1-\epsilon$, we randomly select a new structure from the large search space and add it to the pool if it meets the constraints. The random search is repeated up to $T_{\max}$ times per client sampling. In the worst cases, the smallest model will be chosen to meet current sampling requirements.

FedRolex (Alam et al., 2022) introduces a rolling mechanism to achieve a more balanced parameter selection scheme. In our proposed solution, we can mitigate the issue of unbalanced parameter updates as well. Our random search scheme ensures equal opportunity for selecting all parameters. Each layer has a chance to be individually selected, and all parameters on that layer will undergo training.

### 3.3 Federated In-place Distillation

To further improve the performance of the global model, we propose an in-place distillation process specific to federated learning. Different from previous distillation methods, we do not need public or generated data but can incorporate in-place distillation processes during federated learning. The process of in-place distillation is shown in Algorithm 1. The derivation of Equation (2) in Algorithm 1 is in Appendix B. After the global model is aggregated from the client models, on the server we randomly sample $n$ subnets of different structures from the global model (not from $\mathbb{P}$). To make sure no privacy information of clients is leaked to the server. We let

---

**Algorithm 1:** Federated in-place distillation

---

**Input:** The global model $(\mathbf{W}, \mathbb{S})$

1 **On Server:** ;
2 Random sample $n$ new subnets from $(\mathbf{W}, \mathbb{S})$: $\mathbf{W}_1 \ldots \mathbf{W}_n$.;
3 **for** *iterations $\leftarrow$ 1 to $T_{\mathrm{SKD}}$* **do**
4    **for** $i \leftarrow 1$ *to* $n$ **do**
5       Generate $\mathbb{X}_{\mathrm{KD}}$ with $K$ samples.;
6       **for** $(\mathbf{X}_j, \mathbf{Y}_j) \in \mathbb{X}_{\mathrm{KD}}$ **do**
7          Calculate gradient $\nabla \mathcal{L}_{obj}(\mathbf{Y}_j, \mathbf{W}_i(\mathbf{X}_j))$ where $\mathcal{L}_{obj}$ is an objective loss function, e. g. Cross-entropy loss.;
8          Optimizer $\mathbf{W}_i$ with the gradient for one step and the learning rate is $\eta$.;
9    Update the global model $\mathbf{W}$ with gradients:

$$\nabla \mathbf{W} = \frac{1}{n} \sum_{i=1}^{n} \nabla_{\mathbf{W}^{t-1}} \mathcal{L}_{\mathrm{KD}}(\mathbf{W}^{t-1}, \mathbf{W}_i; \mathbb{X}_{\mathrm{KD}}) \qquad (2)$$

---

each client normalize its local samples on the local datasets first before federated learning starts. Hence, each client does not leak any information about private data to the server. Then, in each iteration of in-place distillation, we generate $K$ samples from a normal distribution $\mathbb{X}_{\mathrm{KD}} \sim \mathcal{N}(\boldsymbol{x}; 0, 1)$, where $K$ is the batch size. As shown in Figure 5, for each sampled subnet, we train them with the soft labels from the global model using the generated $K$ samples in each iteration. After $T_{\mathrm{SKD}}$ iterations of training, we aggregate the weights of these $n$ subnets into the global model. Our in-place distillation module can be applied to other methods that aggregate models of different structures such as HeteroFL.

### 3.4 Activation Function with Boundary

In our search method, apart from different model widths, we also have different model depths and each layer can also be directly skipped. To solve such issue, we add an extra ReLU6 (Sandler et al., 2018) out of the activation function in the neural network to give a boundary over the activation.

## 4 Evaluation

### 4.1 Experiment Settings

#### 4.1.1 Setup for Federated Learning

To thoroughly compare different solutions, we explore various federated settings, including different total numbers of clients, the number of clients participating in each communication round, local epochs, datasets, and model families. We have a total of 6 settings for comparison to cover a range of system configurations. For datasets, we utilize CIFAR10 (Krizhevsky et al., 2009), FEMNIST (Caldas et al., 2018a), and Shakespeare (Caldas et al., 2018a), including both i.i.d. and non-i.i.d. datasets. Non-i.i.d. datasets for CIFAR10 are generated using a Dirichlet distribution with parameter $\alpha$. FEMNIST and Shakespeare datasets are originally non-i.i.d. For detailed client configurations, refer to Table 1.

FEMNIST needs 3597 clients in total and each client has $226.8 \pm 88.94$ samples. Shakespeare needs 1129 clients and each client has $3743.2 \pm 6212.26$ samples. For CIFAR10, training samples are equally partitioned among clients. For the model families, we have ResNet (He et al., 2016), vision image transformers (Dosovitskiy et al., 2021) for image classification and BERT (Devlin et al., 2019) for next character prediction. For the first two settings, the smallest model is the smallest model defined in the corresponding papers: ResNet18 with channels pruned to 6.25%. ViT-tiny-tiny is the ViT-tiny with only one depth. BERT large is the same

Table 1: The client configurations of experiments during FL. CIFAR10 0.1 means the dirichlet parameter $\alpha = 0.1$.

| Setting | Total clients | Clients participated in each round | Local epochs $\tau$ | Dataset |
|---------|---------------|-----------------------------------|---------------------|---------|
| Setting 1 | 100 | 10 | 5 | CIFAR10 0.1 |
| Setting 2 | 100 | 10 | 1 | CIFAR10 0.1 |
| Setting 3 | 50 | 50 | 1 | CIFAR10 i.i.d. |
| Setting 4 | 100 | 10 | 5 | CIFAR10 0.1 |
| Setting 5 | 3597 | 120 | 5 | FEMNIST |
| Setting 6 | 1128 | 120 | 1 | Shakespeare |

as the model in the paper (Devlin et al., 2019) and BERT-tiny only has 6 layers, hidden dimension 256 and attention head 6. The resolution of the images is 32. The sequence length of the text task is 80.

The first two experiment settings, including the settings of the systems, hyper-parameters, models adopted and the distribution resource-constraints are the same as settings in non-i.i.d. cases over the CIFAR10 dataset in HeteroFL (Diao et al., 2021) and high data heterogeneity cases over the CIFAR10 in FedRolex (Alam et al., 2022) respectively. The next three settings simulate system heterogeneity in the real-world and the last setting directly uses the devices' logs of network and memory. We will give more details about system configurations in Section 4.1.2.

Our batch size is fixed at 64 for all experiments. For resource constrained model search, the early stop parameter $T_{\text{MAX}} = 5$ and $\epsilon = 0.8$. The settings of the rest hyper-parameters are in Appendix E. For the federated in-place distillation, the global model is trained with Adam optimizer with learning rate 0.001, $T_{\text{SKD}} = 100$, and $n$ is set the same as number of clients participated in each round. We evaluate the inference accuracy of the global model on the server.

### 4.1.2 System Configurations

To simulate system heterogeneity in real applications, we sample the memory and bandwidth budgets over each client for every communication round. The distribution of the resource constraints is in Table 2. We cover the usual range of available bandwidth and memory on devices (Guo et al., 2021; Yao et al., 2021b; Wang et al., 2020). The memory and bandwidth are independently sampled.

To have a fair and thorough comparison, we have four different types of configurations. In setting 1 and 2, the simulation configuration is the same as in the previous work. We exactly sample 1/5 of the clients having the available resources corresponding to the first model complexity, and 1/5 of the clients corresponding to the second and so on. In setting 3 and 4, memory and network speed are uniformly sampled from the range. In setting 5, the memory is uniformly sampled from the range. We import the devices' logs of transmission rates into the system, using the Wi-Fi logs shown in Figure 1. During the training process, once we reach the end of the logs, we repeat it and start over from the beginning.

In setting 6, we import the logs of transmission rates in real-world applications into the system. We have 4G/LTE bandwidth logs (van der Hooft et al., 2016) and Wi-Fi logs. The 4G logs are collected through running HTTP/2 applications over public transportation (van der Hooft et al., 2016). The logs are of a duration of about 1 hour, which can cover the training processes. We have memory configurations of 4GB and 8GB, which are common configurations of edge devices (Guo et al., 2021; Yao et al., 2021b).

### 4.1.3 Methods Implementation

For a fair comparison, the implementation of baseline methods follows the same approach as described in their respective papers. In both our methods and the baselines, the sampled model must satisfy both the memory and bandwidth conditions. We construct our search space (**W**, $\mathbb{S}$), with **W** representing the largest model as specified in Table 2. For the first two settings, $\mathbb{S} = \{0, 0.0625, 0.125, 0.25, 0.5, 1\}$. For the third and fourth settings, $\mathbb{S} = \{0, 0.5, 1\}$. For the fifth setting, $\mathbb{S} = \{0, 1\}$. And for the sixth setting, $\mathbb{S} = \{0, 0.5, 1\}$. Based on previous empirical observations, we set the frequency for checking resource budgets at the communication

Table 2: The distribution of resource constraints on devices and the models used.

| Setting | Network Speed (Mb/s) | Memory(GB) | Smallest Client Model | Largest Client Model |
|---------|---------------------|------------|----------------------|---------------------|
| Setting1 | [1, 180] | [1.5, 2] | HeteroFL | ResNet18 |
| Setting2 | [1, 180] | [1.5, 2] | FedRolex | ResNet18 |
| Setting3 | [180, 360] | [2, 3] | ResNet18 | ResNet34 |
| Setting4 | [180, 1K] | [2, 6] | ResNet18 | ResNet152 |
| Setting5 | [30, 100] | [1.5, 2] | ViT-tiny-tiny | ViT-tiny |
| Setting6 | [0,110] | {4,8} | BERT-tiny | BERT-large |

Table 3: The inference accuracy of the global model and wall-clock time spent under different settings. We use 'w/' and 'w/o' to denote methods with and without in-place distillation. 'Largest' and 'Smallest' refer to using the largest and smallest models in the search space as the fixed global model in the FedAvg method.

| | Accuracy (%) | | | | | | Latency (hours) | | | | | |
|---------|-------|-------|-------|-------|-------|-------|-------|-------|-------|-------|-------|-------|
| Settings | 1 | 2 | 3 | 4 | 5 | 6 | 1 | 2 | 3 | 4 | 5 | 6 |
| Largest | 71.91 | 75.74 | 92.14 | 74.51 | 84.39 | 65.38 | 19.04 | 13.58 | 5.67 | 33.18 | 6.79 | 0.62 |
| Smallest | 54.16 | 38.82 | 84.50 | 71.91 | 67.86 | 56.96 | 5.06 | 4.94 | 3.88 | 19.04 | 4.57 | 0.60 |
| HeteroFL | 61.64 | 63.90 | 88.76 | 60.95 | 41.87 | 54.32 | 19.54 | 11.18 | 5.48 | 16.53 | 2.69 | 0.62 |
| FjORD | 66.45 | 33.53 | 88.61 | 47.17 | 24.90 | 11.42 | 22.88 | 8.27 | 63.52 | 30.74 | 33.03 | 0.61 |
| FedRolex | 26.50 | 69.44 | 78.63 | 30.53 | 9.22 | 22.39 | 18.88 | 11.43 | 5.25 | 5.98 | 6.57 | 0.62 |
| FedDropout | 18.73 | 46.64 | 34.01 | 14.99 | 22.64 | 14.43 | 15.01 | 11.53 | 4.24 | 7.71 | 7.58 | 0.31 |
| AnycostFL | 14.87 | 14.96 | 87.75 | 69.08 | 8.55 | 12.51 | 18.18 | 6.35 | 5.39 | 7.57 | 7.53 | 0.74 |
| Ours (w/o) | 69.02 | 70.74 | 91.18 | 73.5 | 81.79 | 61.78 | 17.46 | 11.45 | 4.26 | 16.58 | 5.35 | 0.62 |
| Ours (w/) | **69.77** | **71.87** | **91.44** | **74.74** | **83.67** | **64.32** | 18.62 | 12.04 | 5.88 | 17.76 | 5.69 | 0.64 |

round level rather than at each local iteration level. This ensures that when a client checks in for federated learning, its resource budgets remain stable enough to complete one communication round.

## 4.2 Comparison with Baselines

We conduct the comparison experiments under the six settings and the accuracy of converged global model is shown in Table 3. For the method of FedAvg, we show the results of using the smallest and the biggest models. In the actual system, only FedAvg with the smallest model works with resource-constraints requirements on devices. FjORD is a method combining system-heterogeneous federated learning and knowledge distillation. Our methods outperform existing system-heterogeneous FL methods in all settings with the benefits of adopting various model structures and in-place distillation.

With analyzing results in Table 3, we can find the reasons for performance improvement. First, aggregating models of different structures can improve performance compared to using a uniformed structure. Lack of flexibility in existing methodologies is a significant factor contributing to the unsatisfactory outcomes in certain scenarios. In setting 1 and 2, we use the same settings of resource constraints as previous literature, and our methods present performance improvement. This shows that adopting different pruning policy can contribute to the performance improvement.

Second, different settings of resource constraints can also affect the performance. In setting 3 to 6, the range of resource constraints is broader, and more diverse resource constraints exist across devices. For example, comparing setting 1 and setting 4 where the difference is that we set a more realistic resource constraints scheme, the performance of baseline HeteroFL and FedDropout will drop. For example, performance drops severely in particularly hard settings such as setting 4 and setting 5. In setting 5, in order to meet the resource constraints, previous methods choose the model with the lowest 0.0625 pruning rate. In Figure 1, we can see cases where transmission rates are low is frequent. HeteroFL chooses 41% times the model with pruning rate of 0.0625 and the global model fails to get good performance. For method AnycostFL, comparing setting 3, 4 with setting 1, 2, 5, when resources are more constrained, it is not a good idea to

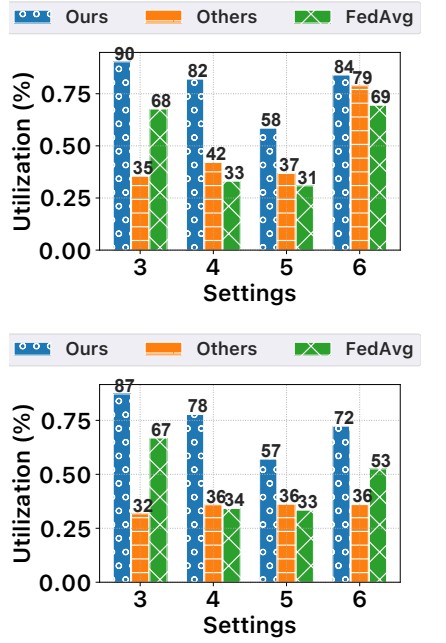

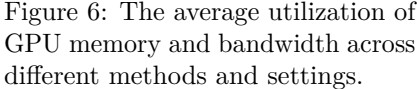

Figure 6: The average utilization of GPU memory and bandwidth across different methods and settings.

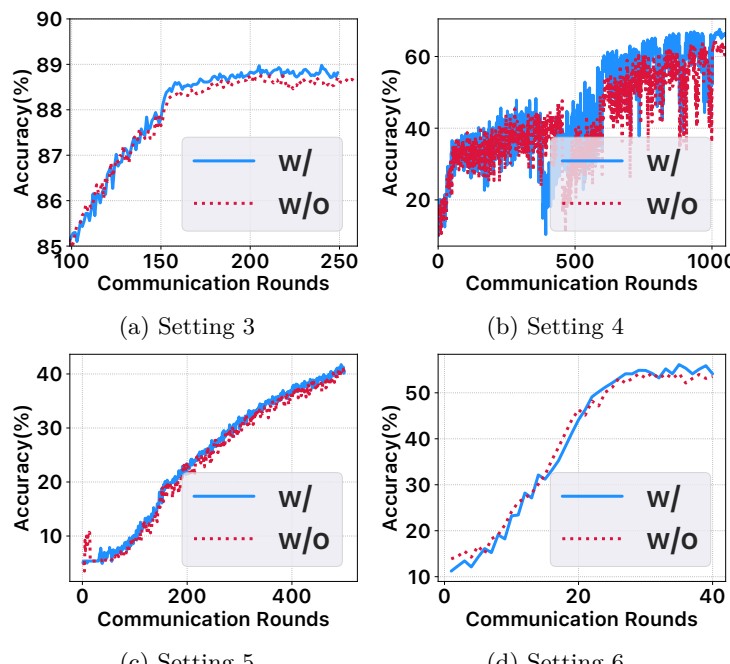

(a) Setting 3      (b) Setting 4

(c) Setting 5      (d) Setting 6

Figure 7: The training curve displays the test accuracy of the global model in HeteroFL with (w/) and without (w/o) in-place distillation across settings 3 to 6.

choose channels based on their L2 norms. Even on a smaller range of the heterogeneity over the resources in setting 1 and setting 2, we still can beat the baselines. To verify this observation, we run an experiment where exactly 1/5 of the clients having the available resources corresponding to the first model complexity and so on. Except for the system configuration, the rest settings are the same as in setting 5. We find that the accuracy of HeteroFL without distillation can be increased to 72.70%, showing that resource constraint can have great impact on the effectiveness of particular algorithms.

Another important metric is how well they fit into clients' system budgets such as the utilization of memory and bandwidth. We measure and compare the utilization of GPU memory and network bandwidth in Figure 6. Because previous methods have the same patterns of choosing only one fixed pruning rate for each model during the training process, they have the same utilization rate. However, this approach often results in sub-optimal model configurations, requiring higher pruning rates to meet both accuracy and utilization requirements, thereby diminishing overall performance. This also implicitly shows that we can find more diverse models. We verify this in setting 3 to 6 where we have more than 5 different cases of resource constraints. In the setting 3, 60 different models are found; in the setting 4, 763 different models are found; in the setting 5, 6 different models are found; in the setting 6, 13 different models are found. If we use the uniform pruning rate, only a maximum of 5 models can be chosen.

### 4.3 Algorithm Overhead

Another important metric to evaluate the proposed method is the extra latency cost. The overhead of our algorithm occurs solely on the server. The clients process the same things as in FedAvg where they receive models from the server and perform training on a single model.

Compared to FedAvg, our server conducts resource-constrained model search and $T_{SKD} \cdot n$ iterations of in-place distillation during each communication round. We need to tradeoff latency for the accuracy improvements. Our server is equipped with an NVIDIA RTX A4500 GPU and 12 CPU cores. Across settings 1 to 5, all resource-constrained model search processes are completed in less than 1 second in each communication round. The total latency on the server, including model search and distillation, in each communication round is 19.65,

15.64, 13.39, 19.16, and 20.5 seconds for settings 1 through 5, respectively. Despite an increase in the number of clients from settings 1 to 5, the overhead does not significantly increase. For setting 1, if we use FedAvg with the smallest model, the latency is 16.87 seconds. This overhead remains unchanged as the number of clients increases, indicating potential scalability. In typical scenarios, the latency of federated learning is primarily determined by communication. Compared to the turnaround time from the server sending models to receiving data from clients, which is usually over 100 seconds, the overhead of our algorithm is negligible. As models become larger, the transmission overhead and computation on the clients increase. Our algorithm's overhead, which is independent of transmission and computation, constitutes a smaller portion of the overall latency and can be neglected.

Comparing the wall-clock latency Table 3, including the time spent on the clients, server, and transmission, we observe that our methods can facilitate faster convergence and achieve higher accuracy. This is because we can maximize the utilization of available resources. Additionally, we can verify that our resource-constrained search adds minimal overhead. When comparing the latency with and without in-place distillation, we observe a 10% increase in overall time, which is acceptable given the improvement in accuracy.

To verify that the criteria of choosing models of the most parameters is a good tradeoff, in setting, we try other criteria. When we choose the model with the largest GPU utilization fitting into the resource requirement, the accuracy is 91.26%. When we use the bandwidth as the criteria, the accuracy is 91.22%. Hence, using the number of parameters can find a better balance point to meet the constraints of memory and bandwidth.

### 4.4 Effects of Search Space Design and Resource Constrained Search

To delve deeper into the design of search space and effectiveness of resource constrained search . We conduct an experiment on setting 3 and the $\mathbb{S}$ of the search space is ($\{1/16, 1/8, 1/4, 1/2, 1\}$), which is the same as the settings of five complexities in our baselines. In this case, we only have search dimensions of different widths on each layer, we still can have benefit comparing to applying one same width pruning rate for all layers, where it can achieve accuracy of 90.04%.

Apart from the search dimension of widths, another search dimension is depth. In the setting 3, we assign a group of experiment that the $\mathbb{S}$ in the search space is {0, 1} and the variance of different structures only lies in different depths. The accuracy is 90.3%. Our method can reach the accuracy very close to the accuracy acquired by the biggest model. In all the settings, even if the whole biggest model is never sampled to the client, we can still compose the biggest model. Taking the chance of small search space, we repeat the experiments by replacing resource constrained search with enumerating all structures, the accuracy is 91.11%. Hence, our resource constrained search can achieve the approximately optimal results.

### 4.5 Effects of In-place Distillation

To evaluate the effectiveness of our proposed federated in-place distillation module. We conduct experiments of removing it. From Table 3, we can see under different settings, in-place distillation module can help improve the final accuracy. Notably, our method can still reach higher accuracy than baselines even without in-place distillation. Different from sBN module in HeteroFL (Diao et al., 2021) and FedRolex (Alam et al., 2022) which may leak privacy as their server needs to query clients about batch normalization statistics, our in-place distillation does not leak privacy.

On the other hand, since our module can be applied to other methods, we implement the process of in-place distillation to HeteroFL and the accuracy and convergence can be improved as shown in Figure 7. The accuracy over HeteroFL can be improved to 88.98%, 67.60%, 41.73%, and 56.13% respectively from setting 3 to setting 6. Regarding FjORD self-distillation module, from Table 3, we can see that it is helpful over i.i.d. datasets (setting 3) but performs poorly over non-i.i.d case (setting 2, 5, 6). Because they conducted distillation over the clients' datasets. Such biased distillation on clients can lead to over-fitting on clients' datasets and poor performance. In contrast, our method can provide performance improvement in both i.i.d and non i.i.d. datasets. Apart from this, though we both share the same extra iterations of knowledge distillation, our in-place distillation is conducted on the server while FjORD is on the clients. As the server usually has more powerful computation ability, our method is more efficient and client-friendly.

## 5 Concluding the Remarks

We revisit the system heterogeneity problem in federated learning by introducing a new system that facilitates algorithm comparison using logs collected from mobile devices. Our focus lies in optimizing models assigned to each client and maximizing resource utilization (memory and bandwidth) on mobile devices. Existing system-heterogeneous methods typically employ channel width pruning with a fixed prune rate for all layers of a neural network, resulting in under-utilization of client resources and poor performance. To tackle this issue, we propose assigning models of various structures to clients, enabling greater flexibility and improved resource utilization. To efficiently sample these models, we introduce resource-constrained model search and federated in-place distillation to enhance performance. Such in-place distillation is applicable to existing system-heterogeneous methods as well. Through our experiments across various settings, we demonstrate the effectiveness of our method, highlighting its potential to enhance system efficiency and client satisfaction.

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

---

**Algorithm 2:** System-heterogeneous federated learning through dynamic model search

---

**Input :** $N$ clients and the search space $(\mathbf{W}, \mathbb{S}), \epsilon, T_{\max}$

**1** Initialize the global model $\mathbf{W}_0$;

**2** Initialize the sampling pool: $\mathbb{P} = \{(s_1, \ldots, s_1), (s_m, \ldots, s_m)\}$;

**3 for** *each round* $t \leftarrow 1$ *to* $T$ **do**

**4**    **On Server:**;

**5**    Sample $\mathbb{N}'$ clients: $\mathbb{N}' \subseteq \{1, \ldots, N\}$;

**6**    **foreach** *client* $i \in \mathcal{N}'$ **do**

**7**       Binary search and sample $\mathbf{V}_i$ from $\mathbb{P}$ with condition $b_i, c_i$;

**8**       loop $\leftarrow 0$;

**9**       **while** *Randomly sample a number uniformly from (0,1) and it is bigger than* $\epsilon$ **do**

**10**          Random search a new $\mathbf{V}'$ from $(\mathbf{W}, \mathbb{S})$;

**11**          $\mathbb{P} \leftarrow \mathbb{P} \cap \{\mathbf{V}'\}$;

**12**          **if** $\mathbf{V}'$ *fits into condition* $b_i, c_i$ *and* $\mathbf{V}'$ *has more parameters than* $\mathbf{V}_i$ **then**

**13**             $\mathbf{V}_i \leftarrow \mathbf{V}_i'$;

**14**          **end**

**15**          loop $\leftarrow$ loop $+ 1$;

**16**          **if** *loop* $> T_{\max}$ **then**

**17**             Break;

**18**          **end**

**19**       **end**

**20**       Sample client model $w_i = w \otimes V_i$ from the global model and then send it to client $i$;

**21**    **end**

**22**    **On Client: foreach** *on client* $i \in \mathbb{N}'$ *parallel* **do**

**23**       Conduct local updates;

**24**       Upload $w_i$ and current constraints $(b_i, c_i)$ to the server;

**25**    **end**

**26**    **On Server:** Receive $\mathbf{W}_i$ and aggregate $\mathbf{W}_i$ to update $\mathbf{W}$;

**27**    Conduct process of inplace distillation;

**28 end**

---

## A   Additional explanation of Overall Algorithm

The whole process of system-heterogeneous federated learning through dynamic model search including the algorithm of resource constrained model search (from line 7 to line 19) is shown in the Algorithm 2.

## B   Additional Explanation about Federated In-place Distillation

We derive Equation (2) from Equation (1). As on the server, we do not have the datasets, we only consider the second part in Equation (1). To update the supernet, we use the gradient:

$$\nabla \mathbf{W} = \nabla_{\mathbf{W}} \mathbb{E}_{\mathbf{W}_i} \mathcal{L}_{\mathrm{KD}}([\mathbf{W}, \mathbf{W}_i]; \mathbf{W}^{t-1}) \tag{3}$$

$$= \frac{1}{n} \sum_{i \in [n]} \nabla_{\mathbf{W}^{t-1}} \mathcal{L}_{\mathrm{KD}}(\mathbf{W}^{t-1}, \mathbf{W}_i; \mathbb{X}_{\mathrm{KD}}) \tag{4}$$

The main idea behind our in-place distillation module is that from Equation (2), we can minimize the distribution divergence between the global model and its subnets through minimizing the KD loss. In such a way, the sampled networks will have a closer output distribution to the global model and so that the aggregation process will be more effective and the performance of the global model will be improved.

During the process of training, the global model is trained in two steps iteratively. It is first trained with aggregation from client models and then trained through in-place distillation. Such a process is repeated

through several communication rounds. So, we can estimate the loss of the global model in federated learning $L_{FL}$:

$$\mathcal{L}_{NAS}(\mathbf{W}) = \mathcal{L}_{\mathbb{D}_c}(\mathbf{W}) + \gamma \mathbb{E}_{\mathbf{W}_s} \mathcal{L}_{\mathrm{KD}}([\mathbf{W}, \mathbf{W}_s]; \mathbf{W}^{t-1}) \tag{5}$$

$$\approx \mathcal{L}_{FL} = \eta \frac{1}{N} \sum_{i \in [N]} \mathbb{E}_{(\boldsymbol{x}, \boldsymbol{y}) \sim D_i} \left[ \mathcal{L}_{\mathrm{obj}}(\boldsymbol{y}, \mathbf{W}_i(\boldsymbol{x})) \right] \tag{6}$$

$$+ \gamma \frac{1}{n} \sum_{i \in [n]} \mathbb{E}_{(\boldsymbol{x}, \boldsymbol{y}) \sim \mathcal{N}(\boldsymbol{x}, \boldsymbol{y}; \boldsymbol{\mu}_{\mathbb{D}}, \sum_{\mathbb{D}})} \left[ D_{\mathrm{KL}}(\mathbf{W}^{t-1}(\boldsymbol{x}) \| \mathbf{W}_i(\boldsymbol{x})) \right] \tag{7}$$

where $\eta$ and $\gamma$ are estimated learning rates for these two steps. The first part of the loss function is an estimation of the expectation and is actually realized through aggregation of client models as the global model is not really trained. The global model is bounded by the objective of federated learning and inplace distillation to minimize the distribution drifts between different clients. As a result, our global model can be viewed as optimized through two targets and thus the performance can be improved.

## C  Proof of Convergence Analysis

Here we give the proof of Theorem 3.3. The high-level idea is to prove that aggregating models of different structures has the same expectation of convergence rate as the conventional federated learning method FedAvg. We first consider the common assumptions in federated learning.

**Assumption C.1.** The loss functions on all clients are all $L$-smooth. We notate them as $F_1, \ldots, F_N'$. For all client models $v_i$ and the global model $W$, we have

$$F_k(v_i) \leq F_k(W) + \nabla F_k(W)^\top (v_i - W) + \frac{L}{2} \|v_i - W\|^2, \forall k \in [N']$$

**Assumption C.2.** The loss functions on all clients are all $\mu$-strongly convex. We notate them as $F_1, \ldots, F_N'$. For all client models $upsilon_i$ and the global model $W$, we have

$$F_k(v_i) \geq F_k(W) + \nabla F_k(W)^\top (v_i - W) + \frac{\mu}{2} \|v_i - W\|^2, \forall k \in [N']$$

.

Assumption C.1 and Assumption C.2 are standard assumptions in federated learning for problems including $l_2$-norm regression and softmax classifier.

**Assumption C.3.** Let $\xi_i$ be sampled from the client $i$'s local data at random. The variance of stochastic gradients in each client is bounded:

$$\mathbb{E} \left[ \|\nabla F_i(\xi_i, v_i) - \nabla F_i(\xi_i, W)\|^2 \right] \leq \sigma_i^2, \forall i \in [N'] \tag{8}$$

**Assumption C.4.** The expected sqaured norm of stochanistc gradients is bounded:

$$\mathbb{E} \left[ \|\nabla F_i(\xi_i, v_i)\|^2 \right] \leq \frac{G}{\tau}, \forall i \in [N'] \tag{9}$$

where $\tau$ is the number of local epochs.

**Lemma C.5.** *In the FedAvg:*

$$\mathbb{E} \left[ \sum_{i \in N'} p_i \|\mathbf{W}^t - v_i^t\|^2 \right] \leq G \tag{10}$$

*where $G$ is the boundary, $v_i$ are client models which has the same structure as the global in FedAvg and $p_i = \frac{|\mathbb{D}_i|}{\sum_{j \in [N']} |\mathbb{D}_j|}$.*

*Proof.* (Proof of Lemma C.5) Start from Equation (9), we assume we have $\tau$ epochs in the communication round $t$. We have

$$\mathbb{E}\left[\|\nabla F_i^j(\xi_i, v_i)\|^2\right] \leq \frac{G}{\tau}$$

$$\sum_{j \in [\tau]} \mathbb{E}\left[\|\nabla F_i^j(\xi_i, v_i)\|^2\right] \leq G$$

$$\mathbb{E}\left[\sum \|\nabla F_i^j(\xi_i, v_i)\|^2\right] \leq G \qquad (11)$$

$$\mathbb{E}\left[\|\sum \nabla F_i^j(\xi_i, v_i)\|^2\right] \leq G$$

$$\mathbb{E}\left[\|\mathbf{W}_i - v_i\|^2\right] \leq G$$

Now, we also take the $p_i$ into consideration, we can have

$$\mathbb{E}\left[\sum_{i \in N'} p_i \|\mathbf{W}^t - v_i^t\|^2\right] \leq \mathbb{E}\left[\sum_{i \in N'} p_i G\right] \qquad (12)$$

$$= G$$

$\square$

Lemma C.5 reveals that for each communication round, the updates over the global model $\mathbf{W}$ is bounded by $G$. This shows that each communication round can result in a bounded step towards the optimal solution $W*$.

**Lemma C.6.** *In federated learning, if Assumption C.1-Assumption C.4 hold and the for each communication round, the updates over the global model $\mathbf{W}$ is bounded by $G$, the convergence rate is $\mathcal{O}(\frac{1}{T})$.*

*Proof.* (Proof of Lemma C.6). We first define several notations. $L$ is defined in *Assumption C*.1, $\mu$ is defined in *Assumption C*.2, $\sigma$ is defined in *Assumption C*.3, $G$ and $\tau$ are defined in Assumption C.4. Let $\kappa = \frac{L}{\mu}$, $\rho = \max\{8\kappa, \tau\}$, and the learning rate $\eta = \frac{2}{\mu(\rho+1)}$. We use $\mathbf{W}^t$ to represent the global model at round $t$ and $\zeta$ to represent the estimated gradient for one step.

$$\|\mathbf{W}^{t+1} - \mathbf{W}^*\|^2 = \|\mathbf{W}^{t+1} - \zeta^{t+1} + \zeta^{t+1} - \mathbf{W}^*\|^2$$

$$= \underbrace{\|\mathbf{W}^{t+1} - \zeta^{t+1}\|^2}_{A_1} + \underbrace{\|\zeta^{t+1} - \mathbf{W}^*\|^2}_{A_2} + \underbrace{2\langle \mathbf{W}^{t+1} - \zeta^{t+1}, \zeta^{t+1} - \mathbf{W}^*\rangle}_{A_3} \qquad (13)$$

When expectation is taken over certain communication rounds, the last term $A_4$ vanishes due to the unbiasedness of $W^{t+1}$. We next calculate the boundary of $A_1$ and $A_2$.

**Lemma C.7.**

$$\mathbb{E}\|\zeta^{t+1} - \mathbf{W}^*\|^2 \leq (1 - \eta\mu)\mathbb{E}\|\mathbf{W}^t - \mathbf{W}^*\|^2 + \eta^2 \sum_{i=1}^{N'} p_i^2 \sigma_i^2 + \eta^2 6L(F^* - \sum_{i=1}^{N'} p_i F_i^*) + 2G \qquad (14)$$

*Proof.* (Proof of Lemma C.7). We first show that

$$\mathbb{E}\|\zeta^{t+1} - \mathbf{W}^*\|^2 \leq (1 - \eta\mu)\mathbb{E}\|\mathbf{W}^t - \mathbf{W}^*\|^2 + \eta^2 \sum_{i=1}^{N'} \mathbb{E}\|g_t - \bar{g}_t\|^2$$

$$+ \eta^2 6L(F^* - \sum_{i=1}^{N'} p_i F_i^*) + 2\mathbb{E}\left[\sum_{i \in N'} p_i \|\mathbf{W}^t - v_i^t\|^2\right] \qquad (15)$$

where we define $g_t = \sum_{i=1}^{N'} p_i \nabla F_i(\xi_i^t, \upsilon_i^t)$ and $\bar{g}^t = \sum_{i=1}^{N'} p_i \nabla F_i(\upsilon_i)$. Therefore $\zeta_t = \mathbf{W}^t - \eta g_t$ and $\mathbb{E} g^t = \bar{g}^t$.

$$
\begin{aligned}
\mathbb{E}\|\zeta^{t+1} - \mathbf{W}^*\|^2 &= \|\mathbf{W}_t - \eta g^t - \mathbf{W}^* - \eta \bar{g}^t + \eta \bar{g}^t\|^2 \\
&= \underbrace{\|\mathbf{W}^t - \mathbf{W}^* - \eta \bar{g}^t\|^2}_{B_1} + \underbrace{2\eta \langle \mathbf{W}^t - \mathbf{W}^* - \eta \bar{g}^t, \bar{g}^t - g^t \rangle}_{B_2} + \eta^2 |g_t - \bar{g}_t\|^2
\end{aligned} \tag{16}
$$

As $\mathbb{E} B_2 = 0$, we can focus on the $B_1$, and we have

$$
\|\mathbf{W}^t - \mathbf{W}^* - \eta \bar{g}^t\|^2 = \|\mathbf{W}^t - \mathbf{W}^*\|^2 - 2\eta \langle \mathbf{W}^t - \mathbf{W}^*, \bar{g}^t \rangle + \eta^2 \|\bar{g}^t\|^2 \tag{17}
$$

From the $L$-smoothness of $F_i$ and convexity of the $L_2$ norm, we have

$$
\eta^2 \|\bar{g}^t\|^2 \leq \eta^2 \sum_{i=1}^{N'} p_i \|\nabla F_i(\upsilon_i^t)\|^2 \leq 2L\eta^2 \sum_{i=1}^{N'} p_i (F_i(\upsilon_i^t) - F_i^*) \tag{18}
$$

We also have

$$
\begin{aligned}
-2\eta \langle \mathbf{W}^t - \mathbf{W}^*, \bar{g}^t \rangle &= -2\eta \sum_{i=1}^{N'} p_i \langle \mathbf{W}^t - \mathbf{W}^*, \nabla F_i(\upsilon_i^t) \rangle \\
&= -2\eta \sum_{i=1}^{N'} p_i \langle \mathbf{W}^t - \upsilon_i, \nabla F_i(\upsilon_i^t) \rangle - 2\eta \sum_{i=1}^{N'} p_i \langle \upsilon_i - \mathbf{W}^*, \nabla F_i(\upsilon_i^t) \rangle
\end{aligned} \tag{19}
$$

We can use the Cuachy-Schwarz inequality and AM-GM inequality to get

$$
-2\langle \mathbf{W}^t - \upsilon_i^t, \nabla F_i(\upsilon_i^t) \rangle \leq \frac{1}{\eta} \|\mathbf{W}^t - \upsilon_i^t\|^2 + \eta \|\nabla F_i(\upsilon_i^t)\|^2 \tag{20}
$$

By the $\mu$-strong convexity of $F_i$, we have

$$
-\langle \upsilon_i^t - \mathbf{W}^*, \nabla F_i(\upsilon_i^t) \rangle \leq -(F_i(\upsilon_i^t) - F_i^*) - \frac{\mu}{2} \|\upsilon_i^t - \mathbf{W}^*\|^2 \tag{21}
$$

Now, we combine Equations (17) to (21) to get

$$
\begin{aligned}
\|\mathbf{W}^t - \mathbf{W}^* - \eta \bar{g}^t\|^2 \leq {}& \|\mathbf{W}^t - \mathbf{W}^*\|^2 + 2L\eta^2 \sum_{i=1}^{N'} p_i (F_i(\upsilon_i^t) - F_i^*) \\
& + \eta \sum_{i-1}^{N'} p_i \left( \frac{1}{\eta} \|\mathbf{W}^t - \upsilon_i^t\|^2 + \eta \|\nabla F_i(\upsilon_i^t)\|^2 \right) \\
& - 2\eta \sum_{i=1}^{N'} p_i \left( F_i(\upsilon_i^t) - F_i^* + \frac{\mu}{2} \|\upsilon_i^t - \mathbf{W}^*\|^2 \right) \\
= {}& (1 - \mu\eta) \|\mathbf{W}^t - \mathbf{W}^*\|^2 + \sum_{i=1}^{N'} p_i \|\mathbf{W}^t - \upsilon_i^t\|^2 \\
& + 4L\eta^2 \sum_{i=1}^{N'} p_i (F_i(\upsilon_i^t) - F_i^*) - 2\eta \sum_{i=1}^{N'} p_i \left( F_i(\upsilon_i^t) - F_i^* \right)
\end{aligned} \tag{22}
$$

We have

$$4L\eta^2 \sum_{i=1}^{N'} p_i(F_i(v_i^t) - F_i^*) - 2\eta \sum_{i=1}^{N'} p_i\left(F_i(v_i^t) - F_i^*\right) = -2\eta(1 - 2L\eta)\sum_{i=1}^{N'} p_i(F_i(v_i^t) - F_i^*) + 2\eta \sum_{i=1}^{N'} p_i(F_i(v_i^t) - F_i^*)$$

$$= -2\eta(1 - 2L\eta)\sum_{i=1}^{N'} p_i(F_i(v_i^t) - F^*)$$

$$+ \eta^2 4L(F^* - \sum_{i=1}^{N'} p_i F_i^*)$$

(23)

And we have

$$\sum_{i=1}^{N'} p_i(F_i(v_i^t) - F^*) = \sum_{i=1}^{N'} p_i(F_i(v_i^t) - F_i^{(}\mathbf{W}^t)) + \sum_{i=1}^{N'} p_i(F_i(\mathbf{W}^t) - F*)$$

$$\geq \sum_{i=1}^{N'} p_i\langle \nabla F_i(\mathbf{W}^t), v_i^t - \mathbf{W}^t \rangle + (F(\mathbf{W}^t) - F^*) \text{ (convexity of } F_i)$$

$$\geq -\frac{1}{2}\sum_{i=1}^{N'} p_i\left[\eta\|\nabla F_i(\mathbf{W}^t)\|^2 + \frac{1}{\eta}\|v_i^t - \mathbf{W}^t\|^2\right] + (F(\mathbf{W}^t) - F^*) \text{ (AM} - \text{GM inequality)}$$

$$\geq -\sum_{i=1}^{N'} p_i\left[\eta L(F_i(\mathbf{W}^t) - F_i^*) + \frac{1}{2\eta}\|v_i^t - \mathbf{W}^t\|^2\right] + (F(\mathbf{W}^t) - F^*) \text{ (}L-\text{smoothness of } F_i)$$

(24)

Now, we put Equation (24) into Equation (23) and we will have

$$4L\eta^2 \sum_{i=1}^{N'} p_i(F_i(v_i^t) - F_i^*) - 2\eta \sum_{i=1}^{N'} p_i\left(F_i(v_i^t) - F_i^*\right) \leq 2\eta(1 - 2L\eta)\sum_{i=1}^{N'} p_i\left[\eta L(F_i(\mathbf{W}^t) - F_i^*) + \frac{1}{2\eta}\|v_i^t - \mathbf{W}^t\|^2\right]$$

$$- 2\eta(1 - 2L\eta)(F(\mathbf{W}^t) - F^*) + \eta^2 4L(F^* - \sum_{i=1}^{N'} p_i F_i^*)$$

$$= 2\eta(1 - 2L\eta)(\eta L - 1)\sum_{i=1}^{N'} p_i(F_i(\mathbf{W}^t) - F_i^*)$$

$$+ (\eta^2 4L + 2\eta^2 L(1 - 2L\eta))(F^* - \sum_{i=1}^{N'} p_i F_i^*)$$

$$+ (1 - 2L\eta)\sum_{i=1}^{N'} p_i\|v_i^t - \mathbf{W}^t\|^2$$

$$\leq \eta^2 6L(F^* - \sum_{i=1}^{N'} p_i F_i^*) + \sum_{i \in N'} p_i\|\mathbf{W}^t - v_i^t\|^2$$

(25)

We plug Equation (25) into the Equation (22) and then plug Equation (22) into Equation (16). We then calculate the expectation on the both sides and we will get the formulation of Equation (15). Combining the Lemma C.5 and Equation (15), we have

$$\mathbb{E}\|\zeta^{t+1} - \mathbf{W}^*\|^2 \leq (1 - \eta\mu)\mathbb{E}\|\mathbf{W}^t - \mathbf{W}^*\|^2 + \eta^2 \sum_{i=1}^{N'} p_i^2 \mathbb{E}\|g_t - \bar{g}_t\|^2 + \eta^2 6L(F^* - \sum_{i=1}^{N'} p_i F_i^*) + 2G \quad (26)$$

We have

$$
\begin{aligned}
\mathbb{E}\|g_t - \bar{g}_t\|^2 = \mathbb{E}\|\sum_{i=1}^{N'} p_i \nabla F_i(\xi_i, \upsilon_i^t) - \nabla F_i(\upsilon_i)\|^2 \\
= \sum_{i=1}^{N'} p_i^2 \mathbb{E}\|\nabla F_i(\xi_i, \upsilon_i^t) - \nabla F_i(\upsilon_i)\|^2 \\
\leq \sum_{i=1}^{N'} p_i^2 \sigma_i^2
\end{aligned}
\tag{27}
$$

The Equation (14) is proved. □

For simplicity, we write Lemma C.7 as

$$
\mathbb{E}\|\zeta^{t+1} - \mathbf{W}^*\|^2 \leq (1 - \eta\mu)\mathbb{E}\|\mathbf{W}^t - \mathbf{W}^*\|^2 + B
\tag{28}
$$

**Lemma C.8.** *The expected divergence between $\zeta^t$ and $\mathbf{W}^*$ is bounded by*

$$
\mathbb{E}\|\zeta^t - \mathbf{W}^*\|^2 \leq C
\tag{29}
$$

*where*

$$
C = \frac{4}{N}\eta^2\tau^2 G
$$

.

*Proof.* (Proof of Lemma C.8). We have

$$
\begin{aligned}
\mathbb{E}\|\zeta^t - \mathbf{W}^*\|^2 =& \frac{1}{N}\sum_{i=1}^{N'} p_i \|\upsilon_i^t - \mathbf{W}*\|^2 \\
=& \frac{1}{N}\sum_{i=1}^{N'} p_i \|(\upsilon_i^t - \mathbf{W}_0) - (\mathbf{W}* - \mathbf{W}_0)\|^2 \\
\leq& \frac{1}{N}\sum_{i=1}^{N'} p_i \|(\upsilon_i^t - \mathbf{W}_0)\|^2 \\
\leq& \frac{1}{N}\sum_{i=1}^{N'} p_i\tau \sum_{j=0}^{t} \mathbb{E}\|\eta\nabla F_i(\xi_i^t, \upsilon_i^t)\|^2 \\
\leq& \frac{1}{N}\tau^2\eta^2 G
\end{aligned}
\tag{30}
$$

□

According to Lemma C.7 and Lemma C.8, we have

$$
\mathbb{E}\|\mathbf{W}^{t+1} - \mathbf{W}^*\|^2 \leq (1 - \eta\mu)\mathbb{E}\|\mathbf{W}^t - \mathbf{W}^*\|^2 + B + C
\tag{31}
$$

For a diminishing stepsize, $\eta = \frac{\beta}{t+\rho}$ for some $\beta > \frac{1}{\mu}$ and $\rho > 0$ such that $\eta \leq \frac{1}{4L}$. We next prove that $\Delta^t = \mathbb{E}\|\mathbf{W}^{t+1} - \mathbf{W}^*\|^2 \leq \frac{\nu}{\rho+t}$ where $\nu = \max\{\frac{(t+\rho)^2(B+C)}{\beta\mu-1}, (\rho+1)\Delta^1\}$. We prove this by induction. First, th definition of $\nu$ ensures that it holds for $t = 1$. We assume it holds for some $t$. We will have

$$
\begin{aligned}
\Delta^{t+1} &\leq (1 - \eta\mu)\Delta^t + B + C \\
&\leq (1 - \frac{\beta}{t+\rho}\mu)\frac{\nu}{\rho+t} + B + C \\
&= \frac{t+\rho-1}{t+\rho}^2\nu + \left[B + C - \frac{\beta\mu-1}{(t+\rho)^2}\nu\right] \\
&\leq \frac{\nu}{\rho+t+1}
\end{aligned}
\tag{32}
$$

Then by the $L$-smoothness of $F$, we have

$$\mathbb{E}[F(\mathbf{W}^t)] - F^* \leq \frac{L}{2}\Delta^t \leq \frac{L\nu}{2(\rho + t)} \tag{33}$$

Sepcifically, if we choose $\beta = \frac{2}{\mu}$, we can have

$$\mathbb{E}[F(\mathbf{W}^t)] - F^* \leq \frac{L(\rho + 1)}{2(\rho + T)}\left((1 + \rho)(B + C) + \|\mathbf{W}_0 - \mathbf{W}^*\|^2\right) \tag{34}$$

Therefore, with a given communication round $T$, the difference between the accuracy achieved by the global model and the optimal accuracy is bounded by the $O(\frac{1}{T})$. $\qquad\square$

With Lemma C.5 and Lemma C.6, the convergence rate of FedAvg is $O(\frac{1}{T})$. As a result, to prove that the convergence rate of aggregating models of different structures is also $O(\frac{1}{T})$, we need to show that the updates over the global model $W$ in our method is also bounded by $G$. We will prove this in the following lemma.

**Lemma C.9.** *In the system-heterogenous federated learning, we use $\mathbf{W}^t/\mathbf{W}_i^t$ to represent the difference between the global model and the subnet. We have*

$$\mathbb{E}\left[\sum_{i \in N'} p_i \|\mathbf{W}^t/\mathbf{W}_i^t\|^2\right] \leq G \tag{35}$$

*Proof.* (Proof of Lemma C.9) During the model search step, we sample the $\mathbf{W}_i$ which is a subnet from the global model. For the parameters which are not sampled in $\mathbf{W}_i$, they will be 0. As a result, we can construct a virtual model $\theta_i^t$. This virtual model has the same structure as the $\mathbf{W}^t$ but the values of the parameters are different. The parameters should be the same as those in $\mathbf{W}_i$ if they are not 0. For the rest parameters, we can assign them particular values to make them have the same distribution as $\upsilon_i^t$. We will have

$$\mathbb{E}\left[\sum_{i \in N'} p_i \|\mathbf{W}^t/\mathbf{W}_i^t\|^2\right] \leq \mathbb{E}\left[\sum_{i \in N'} p_i \|\mathbf{W}^t - \theta_i^t\|^2\right] \leq G \tag{36}$$

$\square$

*Proof.* (Proof of Theorem 3.3) According to Lemma C.6, and Lemma C.9, the convergence rate of aggregating models of different structures is also $O(\frac{1}{T})$. $\qquad\square$

# D System Design

We implement our system based on PLATO (Li et al., 2023a), an open-source framework for FL. PLATO provides interfaces easy of use, which can simulate cross-device scenarios with a lot of devices and various data distribution. Plato is compatible with existing FL frameworks and infrastructures, providing a good development base for our system.

## D.1 System Requirements

*First*, the system should **accurately simulate the available resources in real-world scenarios**. In previous simulation processes, such as those used in HeteroFL and FedRolex, models of different sizes are equally assigned to clients. For example, with 10 clients per round, each client receives models of varying complexities. It is assumed that there will be precisely two devices capable of running each complexity level, ensuring a balanced distribution. Additionally, 1/5 of all clients are expected to qualify for each of the five designated models in every round.

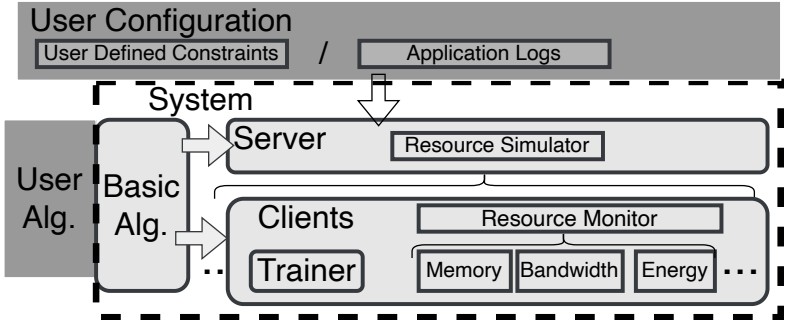

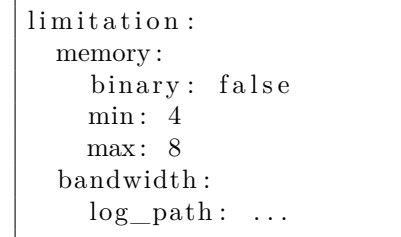

Figure 9: Configuration.

Figure 8: System structure.

*Second*, the system should **measures directly in metrics evaluating resources (memory, bandwidth) rather than proxy metrics (FLOPS, number of parameters).** Our interfaces facilitate accurate assessment of memory and bandwidth usage. When we implement our proposed algorithm, we search for model structures directly through its constraints.

*Thirdly*, **reproducibility** is crucial for establishing a fair comparison. With the users' provided system configurations, we can compare various algorithms under the same settings and replicate the experiments..

## D.2 System Implementation

We build the functionalities of our system in four modules *Server*, *Client*, *Trainer*, and *Algorithm* based on PLATO interfaces. The overall structure is shown in Figure 8. We have a server to load in user configuration. The server will correspondingly assign the available memory, bandwidth, etc. to clients and the clients will monitor the usage of these resources. We implement different system-heterogeneous algorithms in the user algorithm part, on top of the basic algorithm (FedAvg). In such a way, we can control the only variable: algorithms used, during experiments.

Notably, our design of separating modules can accurately simulate the wall-clock elapsed time in federated learning. For example, it accounts for the time spent running algorithms and training models in a trainer but excludes overhead spent on simulating the system, such as time spent in resource simulation and monitoring.

### D.2.1 Server Module

On the server, we will import the user configurations and the server will correspondingly assign the resource budgets. We show the example configurations in Figure 9. In the *limitation* parameter, users will first list out the types of resources, for example, memory, bandwidth, energy. Under each kind of resource, users have two choices. They can either assign a maximum and minimal value or assign a path to load logs. If users choose the former, the budget will be uniformly sampled from the corresponding range. In the former option, if the user sets binary to true, the value will be only sampled from the two values (maximum and minimum). If they choose the latter, the server will import logs of each client device. Each log file should log the timestamps and corresponding values. In PLATO, as the wall clock elapsed time is provided, we can quickly search the corresponding resource budget at the timestamp we need to assign models.

We also provide an interface for checking the resource usage of a given model structure in the server. When the algorithm module searches for the proper model structure, the server will evaluate how much memory it will use or how much time it will take to transmit the model according to the current set of resource budgets.

### D.2.2 Client Module

The client module consists of a resource monitor and a *Trainer* module. The trainer receives the assigned model and trains it with local data. The resource monitor tracks current resource utilization; if the current utilization exceeds the assigned budget, the client will throw corresponding errors. Additionally, we may encounter cases where the available budgets change during local training, though this occurs infrequently.

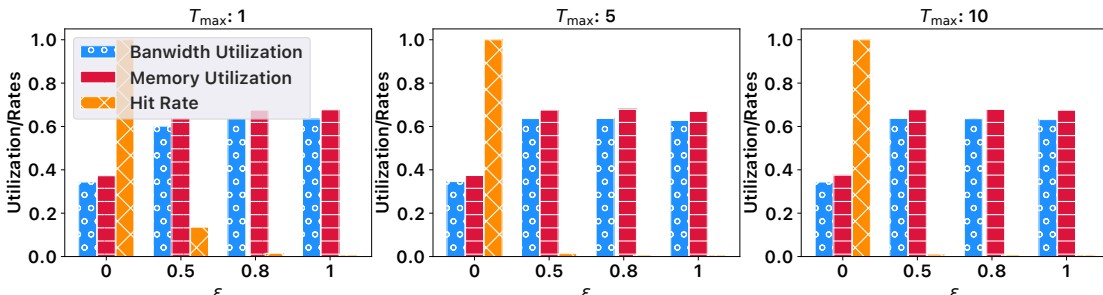

Figure 10: The utilization of bandwidth and memory as well as hit rates with different $\epsilon$ and $T_{\max}$ during resource constrained model search.

Therefore, we upload the client logs to the server module. The server module compares the client logs, local training time, and logging file in the user configuration. If the resource exceeds the budget during client training, the server treats this time as a failure in training on the client.

## E    Hyper-parameter Settings

Here we show the rest settings of the hyper-parameters during experiments. For the settings 1, 3, and 4, the hyper-parameters are the same as the parameters used in HeteroFL. For setting 2, they are the same as those in FedRolex. For setting 5, they are the same as hyper-parameters in training ViT on ImageNet (Dosovitskiy et al., 2021). For setting 6, they are the same as hyper-parameters in training BERT (Devlin et al., 2019). For the setting 1,3, 4 and 6, we use the SGD optimizer with learning rate 0.1, momentum 0.9 and weight decay 0.0005. For the setting 2, the learning rate is 0.0002 and the rests are the same. For the setting 5, we use AdamW optimizer with learning rate 0.001, $\beta$s are $(0.9, 0.99)$ and weight decay is 0.01. We also use the learning rate scheduler, for the setting 1 to 4 and 6, we use the multi-step scheduler with a decay rate of 0.1. For setting 1, we decay at round 300 and 500. For setting 2, we decay at round 800 and 1250. For setting 3, we decay at 150 and 250. For setting 4, we decay at 300 and 500. For the setting 6, we decay at 10 and 20. For setting 5, we use the cosine scheduler and the cycle is 500 rounds.

### E.1    Hyper-parameters in Resource Constrained Model Search

During the resource constrained model search, there are two important hyper-parameters: $T_{\max}$ and $\epsilon$. Intuitively, larger $\epsilon$ will lead to less exploration, which means less possibility to find proper structure. On the other hand, more exploration involves more random search, which means possible more running time. Larger $T_{\max}$ will increase the possibility of expanding the sampling pool and find more suitable structures but can lead to increasing running time. Before start the process of federated learning to measure the accuracy and efficiency, we do an experiment to find out the suitable $\epsilon$ and $T_{\max}$.

In this experiment, we do not really train each model but only measure the utilization of assigned GPU memory and bandwidth with the given single model on each client. Higher utilization means we find a more suitable model to fit into the given GPU memory and bandwidth and such a model can better help train the global model. We also measure the rate that whether the $T_{\max}$ is reached, which we define as hit rates. We use setting 5, which uses the logs from the real world and simulates for 100 communication rounds. Figure 10 shows the average utilization and hit rates among all participant clients and communication rounds. We can see that lager $\epsilon$ can help get better utilization. To introduce the exploration into the search process, we adopt $\epsilon = 0.8$. We can see that when $T_{\max}$ increases, the hit rate will be smaller. However, too large $T_{\max}$ may lead to worse efficiency. So we adopt $T_{\max} = 5$.

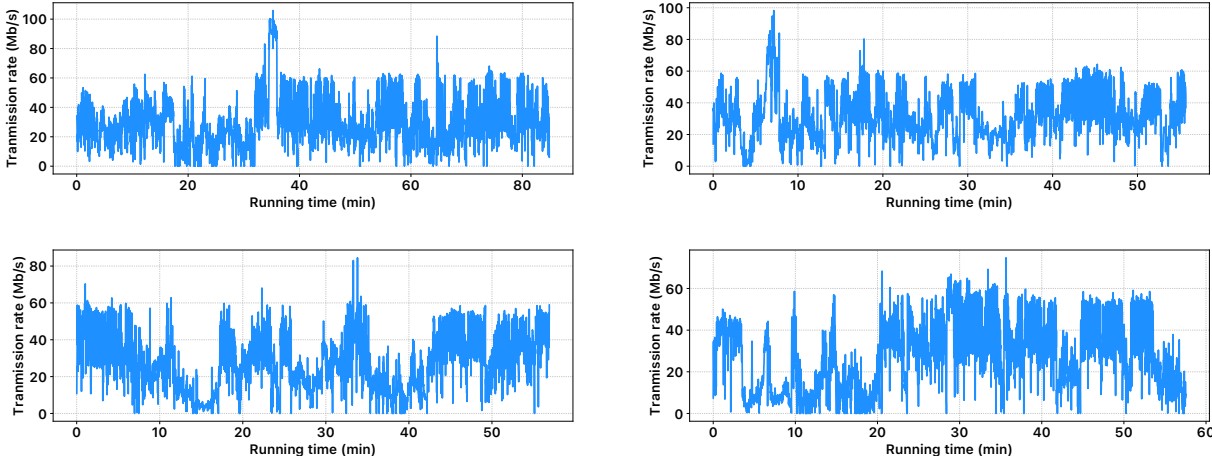

Figure 11: The visualization of 4G/LTE bandwidth logs over different public transportation: Bus, Car, Foot, and Tram

### E.2 Hyper-parameters in In-place Distillation

There are two hyper-parameters we need to set during the in-place distillation. We use the Adam optimizer with learning rate 0.001, which is a common setting of the optimizer. The $T_{\mathrm{SKD}}$ is set as 100. There is no particular value for this hyper-parameter. Any reasonable value is fine. We find that after $T_{\mathrm{SKD}}$, it does not bring much accuracy improvement in the final results. But the latency spent on the server will linearly grow. So we choose the value of 100.

## F    Details of 4G/LTE logs

We show the details of these logs in Figure 11.

## G    Additional experiments

### G.1    Comparison with Baselines

The convergence rate during the training process is shown in Figure 12.

### G.2    Impact of Failing to Meet Resource Constraints

In federated learning, when facing resource constraints, a trivial solution is to employ a smaller model. However, this may result in lower accuracy than what can be achieved with larger models. Given that resource constraints vary among clients in FL, some clients may possess unused resources that could be leveraged to train larger models.

Another solution is to exclude low-resource clients. Nonetheless, if these clients persistently have limited resources, they will never be able to participate in federated learning. We conducted toy experiments (Non IID $\alpha = 0.1$, 10/100) where we have 100 clients and 10 out of them are selected in each round. We construct a non-i.i.d. CIFAR10 (Krizhevsky et al., 2009) dataset with Dirichlet distribution (parameter $\alpha = 0.1$). The local epoch in each round is 5. We will exclude clients if they cannot run the model after client selection. The resources limitation on each client is fixed and will not change between each communication round.

We choose different ResNet models leading to different participation rates. The results as well as the sizes and flops of these models are shown in Figure 13. Lower participation rates will result in low performance because we cannot make use of data on most of the clients. In this case, this solution is even worse than adopting a

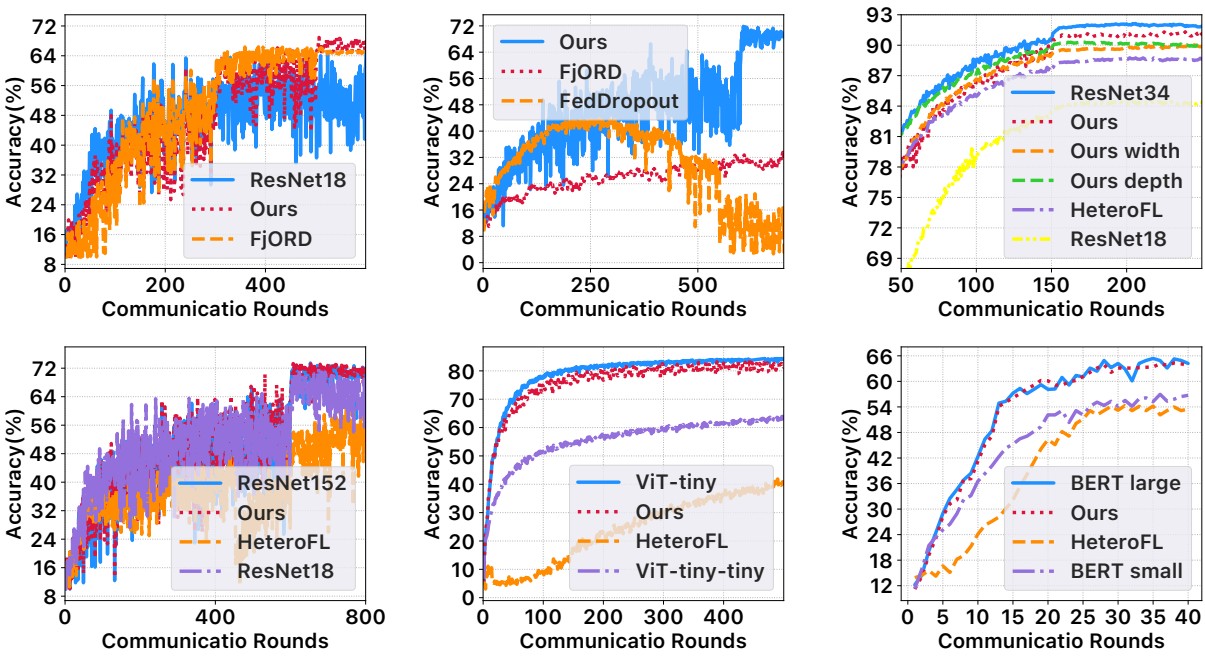

Figure 12: The training curve of the test accuracy of the global model of our methods and the baselines under from setting 1 to setting 6. Ours width/depth means our method with search space only containing different widths/depths.

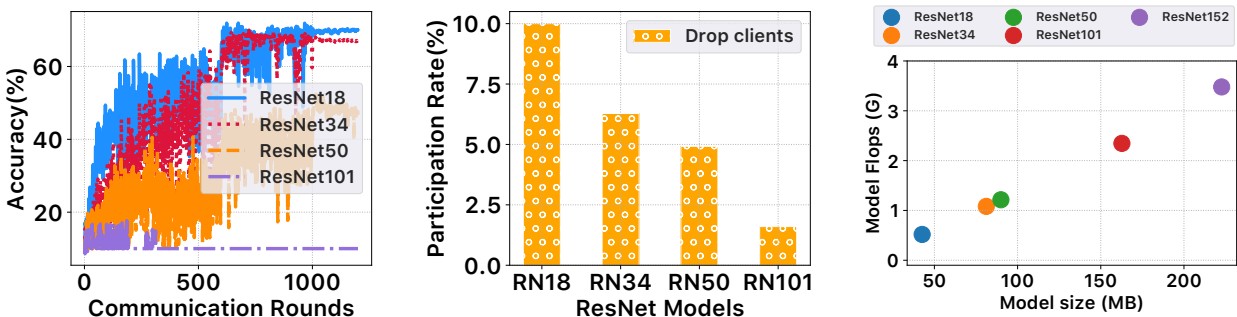

Figure 13: The global model accuracy and client participation of federated learning with clients dropping due to clients' various resource constraints.

smaller model. As a result, we need a resource-aware solution to have a high utilization of available resources, achieving high accuracy.

Rather than adopting uniform models, there are approaches assign heterogeneous models to clients. Numerous existing methods focus on optimizing communication and computation overhead on clients, such as structured pruning, compression, and quantization. However, untargeted optimization cannot solve the hard-constraints problem. For instance, Hermes (Li et al., 2021) uses structured pruning to balance communication and computation efficiency. It prunes channels with the lowest magnitudes in each local model and adjusts the pruning amount based on each local model's test accuracy and its previous pruning amount. But its adaptive pruning rate can still result in client failure to run the model because the prune rate is not calculated directly from the clients' resource constraints. Additionally, at the early stage of the algorithm, the clients and the server must transmit the entire model. Hence, even with the implementation of heterogeneous models, issues in system-heterogeneous FL may not be entirely resolved.

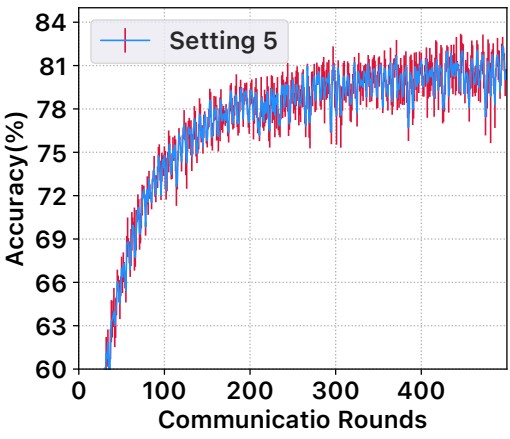
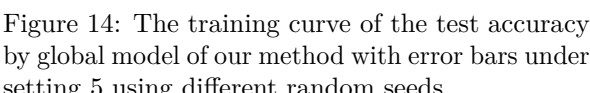

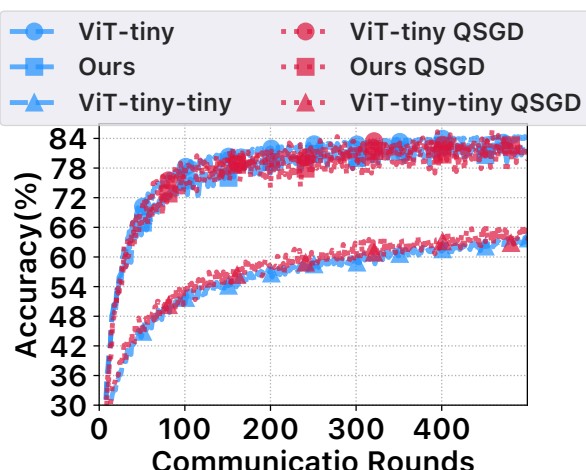

Figure 14: The training curve of the test accuracy by global model of our method with error bars under setting 5 using different random seeds.

Figure 15: The training curve of the test accuracy by global model of our method and FedAvg after applying QSGD.

### G.3 Stability and Convergence

As there are randomization mechanisms in our methods during resource constrained model search and inplace distillation, we also test our methods over different random seeds to see if the randomness will affect the performance of our method. We repeat the experiments of setting5 with setting different seeds. The best inference accuracy of the global model in these three experiments are 83.20%, 83.15%, 83.67%. As shown in Figure 14, the convergence and final accuracy will not be affected by the randomness.

Besides theoretical convergence guarantee and analysis of our randomness effects on the communication round in Section 3.1.3, we further show the convergence through empirical experiments. From the convergence curve in Figure 12, our method can converge at the same rate or even faster than the baselines. In setting 1, our methods only need 311 rounds to reach the accuracy reported by HeteroFL where their setting for communication rounds is 800. In setting 2, our method only needs 1263 rounds to reach the accuracy reported by FedRolex where their setting for communication rounds is 2500. As a result, we are also efficient in total latency.

### G.4 Orthogonal to other Efficiency Methods

We have shown that our methods can better solve resource-aware problems in federated learning compared to existing system heterogeneous methods. There are other conventional distributed learning efficiency methods. Though they cannot resolve system heterogeneous problems, they still can help improve efficiency. Our method is orthogonal to those methods such as compression and quantization. We apply the QSGD Alistarh et al. (2017) which is a compression method to reduce the communication overhead in distributed systems. The quantization level is 64 and we quantize from 32 bits to 8 bits. We apply this process to our methods and FedAvg and the results in shown in Figure 15. Our method still works after applying QSGD. As a result, our method is orthogonal to previous efficiency methods and the combination of such methods with our methods will further help improve system efficiency.

## H Discussion

A limitation of our system is that we mainly develop the system with PYTORCH framework. As a result, the interface for counting CUDA memory utilization are based on the TORCH API. As PLATO also support other deep learning frameworks such as TENSORFLOW and MINDSPORE, we may further extend such interfaces of maintaining resource over other deep learning frameworks.

In this paper, we focus on two types of resources: memory usage and network speed. The reason is that these two kinds of resources may change more frequently than other possible resources during the federated training process, which can emphasize the importance of handing system-heterogeneity during federated learning. During federated learning, devices can also be made of hardware with different computation ability and energy consumption. However, users will not frequently change energy mode during the few hours learning process. The computation ability will not change during federated learning. So we place them in a minor consideration. While on the other hand, our system's modular design allows us to add in other constraints. In our algorithm, we can also add or change the resources with setting different constraints during the implementation of line 8 in Algorithm 2. In short, our system and proposed model search method can have an easy extension to other resources besides memory and bandwidth.

Another direction of future work is about privacy. In this paper, in the perspective of each client, the training process the same as that in FedAvg. In other words, methods which are applicable to conventional federated learning such as differential privacy can be also applicable to our algorithms. Our system is developed upon Plato, where a lot of privacy-preserving methods in federated learning are built-in. To the best of our knowledge, there are no existing attacking methods or privacy leakage attempts targeted on models of different structures in federated learning. Hence, our methods currently can preserve privacy under existing federated learning protocol. As this paper focuses on system level optimization, we leave the research question of finding such an attacking method for future work.

In our system, we have the flexibility of choosing logs for users. Users may need to ensure the accuracy of logs before inputting them into the system. The problem we are trying to resolve is allowing users to compare different system heterogeneity fairly with reproducible logs. Users can use any logs they want, the real-world logs, the logs generated by simulators or even logs under unreal settings. For example, the settings in HeteroFL and FedRolex cannot reflect practical use cases, but we can still compare different methods using their settings under our system fairly.

To resolve the question of how we can utilize available memory and network bandwidth to the maximum, besides system-heterogeneous federated learning, another direction is leveraging split learning. However , split learning requires the clients and the server to communicate at every iteration, which is quite inefficient in communication, compared to federated learning. Apart from that, to ensure privacy is preserved in split-learning, users still need to load certain layers on the clients. In other words, if users put a few layers on the clients to meet resource budgets, privacy may not probably be well preserved.

