# OpenReview forum: "Revisiting System-Heterogeneous Federated Learning through Dynamic Model Search"
_TMLR — Rejected by TMLR_

### Review · Reviewer_6BnQ · 2024-06-16

**Summary Of Contributions:**

This paper proposes a novel training algorithm for System-Heterogeneous Federated Learning. The primary contribution lies in expanding the search space of client models by introducing variable channel pruning ratios across different layers, in contrast to the uniform pruning ratios used in previous works. Additionally, the authors employ a data-free knowledge distillation algorithm for global model training, which does not require public data. They simulate real-world resource configurations and validate the proposed approach on CIFAR-10 scale datasets using diverse architectures, including ResNet and Vision Transformers. The experimental results demonstrate that the proposed search space and distillation algorithm improve previous approaches by large margins.

**Audience:**

Yes

**Broader Impact Concerns:**

They are partially discussed in Appendix Section G. It would be beneficial to provide a more detailed description on a broader impact.

**Claims And Evidence:**

Yes

**Requested Changes:**

- How would the performance change if the client model was selected based on criteria such as GPU utilization, bandwidth, and overall performance, rather than simply the number of parameters? For efficiency, we might adopt subnetwork evaluation techniques from the NAS field.
- It would be beneficial to include qualitative analysis to confirm whether more diverse and effective client models were indeed selected compared to uniform channel pruning.
- In Table 3, the performance of existing methods on the ViT model is significantly lower. Can you confirm what differences in the client models might have led to this performance disparity?

**Strengths And Weaknesses:**

Strengths
- The writing is clear and easy to follow, making the concepts accessible.
- The experiments are conducted thoroughly, simulating real-world resource configurations to validate the approach.
- Ablation studies effectively support the proposed techniques, providing solid evidence of their benefits.

Weaknesses
- The criterion for selecting client models requires further justification and support. (i.e., selecting the client model with the largest parameter size satisfying the resource constraint.)
- The different pruning ratios and data-free knowledge distillation algorithms used are not technically new methods, which limits the novelty of the technical contributions.

---

### Review · Reviewer_sKQA · 2024-06-21

**Summary Of Contributions:**

The paper presents an approach to federated learning, where each edge device may have varying memory and network width, i.e., a more realistic setting. The key approach to do this to use different local models over time; the contribution of the paper are a number of smaller things to do this. The paper also reports on a tool to simulate federated learning in such a setting and reports on an experimental study.

**Audience:**

Yes

**Broader Impact Concerns:**

.

**Claims And Evidence:**

No

**Requested Changes:**

Address W1,W2,W4

**Strengths And Weaknesses:**

S1. More realistic setting than the assumption of constant resource availability

S2. A number of smaller technical contributions: (i) to assign models based on current capacity to each client, (ii) to use different pruning rates for different layers, (iii) to aggregate parameters across the different pruning rates, (iv) and to perform in-place distillation using synthetic data.

S3. Higher global-model accuracy than considered alternative models in experiments

W1. Experimental study lacks insight. Most importantly, the reason for performance improvements remains unclear. I.e., it's not clear whether the performance of the approach is due to handling varying resource constraints or rather due to differences in pruning and aggregation (and if so, which one). Clear and fair ablation studies are needed. Additionally, there is also no analysis of accuracy/latency tradeoffs and accuracy/cost tradeoffs (which is probably very high for the proposed method).

W2. There is no analysis on potential implications of the proposed techniques on quality and privacy. For the former, a "convergence analysis" is presented in Th. 3.3., but not the theorem statement is not clear and the proof is not self-contained. It's also not clear which method has actually been analyzed (e.g., with in-place knowledge distillation). For the latter, a single sentence can be found at the end of the appendix, essentially saying that privacy implications remains unclear.

W3. Low novelty as all technical contributions are smaller tweaks. This would generally ok if it weren't for W1 and W2.

W4. Presentation. The paper is too long for its content, i.e., it uses many words to say simple things. There is also a lack of focus; e.g., tools such as the simulator are probably very helpful, but they shouldn't be described in the main paper. Finally, there is often little justification for why certain techniques haven been applied / are beneficial and what the assumptions are. E.g., in-place distillation seems to use mean/variance parameters of the data. This seems to make a normality assumption, it's not clear how to get the parameters in a privacy-preserving fashion, and it's not clear why all of this is a good idea.

---

### Review · Reviewer_PMrb · 2024-07-03

**Summary Of Contributions:**

I previously post review as an "Official Comment" titled "Interesting but ambiguous ".

**Audience:**

Yes

**Claims And Evidence:**

No

**Requested Changes:**

Thank the authors for their feedback. In this current revision, I still have some confusions.

***1. About the distillation.***
1) Can the authors explain the rationale of applying distillation here in the federated learning setting? It is my understanding that distillation tries to train a smaller model while maintaining the performance, which seems not what this paper is doing.

2) $\mathcal{L}$ has been defined in inconsistent formulations, such as in eq (1), Line 7 of Algorithm 1, and eq (5), eq (6). Please check and consider editing them to make them appear consistent. $L_{KD}$ also has inconsistent formulation in eq (1) and eq (4) since you have a $W_{t-1}$ in eq (1).

3) Line 9 in Algorithm 1: Should it be moved left to match the indentation of Line 4?

4) Line 10 in Algorithm 1: Did you miss a gradient sign on the right hand side of the equation?

***2. About the sub-structure initialization***
The authors did not address my concern of how to initialize a newly sampled sub-structure.

***3. About the convergence analysis***
The analysis in the revision is still depending on a previous literature. But with all the algorithmic design such as sub-structure sampling, initializing a sub-structure when it is sampled, aggregation, and distillation, it might be proper to directly plug in the analysis framework in that literature. I do not think an analysis is always required, but an analysis should be self-contained to support the theoretical claim.

**Strengths And Weaknesses:**

I previously post review as an "Official Comment" titled "Interesting but ambiguous ".

---

> ### Author Response · Authors · 2024-07-04
> **Response to Reviewer PMrb**
>
> Thank you very much for taking to time to review our updated manuscript and provide insightful suggestions.
>
> Regarding the confusions:
>
> ### 1. About the distillation
> 1. The in-place distillation was originally used in centralized NAS. In such a case the distillation is used to improve the accuracy of the super-net where the super-net is aggregated from a lot of sampled sub-networks. In our federated learning setting, we have a similar case where we need to aggregate several sampled sub-networks into a global big network. And we want to improve the accuracy of global big networks. In section 2.3, we explain this reason for why we apply such a kind of in-place distillation into a federated learning setting. We are basically trying to bring the benefits of centralized NAS into federated learning settings.
>
> We are not trying to train smaller models while maintaining performance. But there exists a connection between such an objective and what we are doing in the paper. During in-place distillation, we sample a lot of small models (line 2 in Algorithm 1) and we do in-place distillation to make sure they have the same performance as the global model. Through sampling a certain number of sub-networks during distillation, we make any newly-sampled network has the same performance as the global model. In such a way, we further make sure that in the next run, when clients use the newly-sampled models to do local training, they are doing the local training on models having the same performance as the global model. Then, when we aggregate these trained models into a global model. The global model will have a good performance as it is aggregated from trained models whose performances are not degraded.
>
> 2. $L$ in eq (1), Line 7 of Algorithm 1, and eq (5), eq (6) represent different loss functions respectively. In eq(1), it is the loss function in centralized NAS. In line 7 of Algorithm 1, it is the objective loss function, for example, cross-entropy loss in an image classification task. In eq(5) and eq(6), it is the estimated loss function in federated learning for updating the global model . We revised the manuscript to make this clear. Regarding $L_{KD}$, we revised eq(1) and eq(4) to make it consistent.
> 3. Yes. Thank you. We revised it.
> 4. Yes. Thank you. We added it. We also added the missed gradient sign in equations (3) and (4).
>
> ### 2. About the sub-structure initialization
> According to definition 3.2, we will first search a vector, e.g. $V_1$. For example, in figure 3, on the left, $V_1=(0.5,0.5,0.5,0.5)$ where for each layer the input channel width is a half of the channel width of the global model. For example, if the third layer in the global network is a convolution layer and we use the following hyper-parameters: input channel width of 16, output channel width of 32, kernel size of 3, etc. If $v_3=0.25$, when we initialize a new sub-net, we build a convolution layer with hyper-parameters: input channel of 16, output channel of 8. Kernel size of 3, etc.
>
> Let's use setting 3 as an example, where the largest client model is ResNet34. In ResNet34, we have 3 blocks with hidden widths of 64, 4 with 128, 6 with 256, and 3 with 512.  If in the random search, we choose to cut all widths to the half, we will construct a ResNet with 3 blocks with 32, 4 with 64, 6 with 128, and 3 with 256. And if we choose to prune the last layer, we will have only 2 blocks with hidden widths of 256.
>
> After the model structure is initialized, the model weights will be directly copied from the global model. As the model architecture is not changed, for example, the sub-net and the global model are both ResNet. Only the hyper-parameter such as channel widths and kernel sizes for the corresponding layers change. So, we can copy a part of the parameters from the global model into the sub-nets.
>
> We added the explanations in the section 3.1.1.
>
> Regarding the first two points, we uploaded a new version of the revised manuscript.
>
> ### 3. About the convergence analysis
> Thank you for the suggestion. We will directly plug in the analysis framework in that literature and provide another version of the revised manuscript very soon.

---

> > ### Author Response · Authors · 2024-07-04
> > **Response to Reviewer PMrb**
> >
> > According to the suggestion regarding the convergence analysis, we have plugged in the analysis framework in that literature to make our proof self-contained. Thank you.

---

### Comment · Reviewer_PMrb · 2024-06-19
**Interesting but ambiguous**

This work targets at the issue of heterogeneity of clients' computing resource and bandwidth. The solution is to allow clients to train models of different sizes and aggregate them using distillation. The considered setting is indeed significant, and the proposed algorithms have shown some interesting thought. However, there are some points that appear to be ambiguous to me and I wish the authors could address them in revision.

 1) Looks the models used on different machines have different number of layers and different sizes at each layer, but the models generally have the same framework. That means, if the global model is a ResNet34, the local models are part of the ResNet34. If that is the case, it might be misleading to say different architectures are used in different clients.

2) The algorithms have used some descriptions that are not clear enough for understanding and implementing. To name some, Algorithm 1 Line 6 (how exactly it goes), Algorithm 2 Line 9 (what is the "probability" here, how to update it), Algorithm 2 Line 12 (how do you define "better").

3) How to aggregate different models? Figure 4 is unclear to me. I suggest the authors to clearly describe the aggregation parts, which has only be briefly discussed with a few sentences in the ending of Section 3.1. Besides, what is the connection between aggregation and distillation? Meaning, after aggregating kernels of different sizes together, do you need to do the distillation and if so, how?

4) How to initialize a layer/kernel/architecture when it gets sampled?

5) The convergence analysis, i.e., Theorem 3.3 and its proof, are far from complete and solid. The referred literature Li et al. (2020) did not involve different architectures, thus is questionable whether their results can directly imply Theorem 3.3. I suggest the authors to clearly and fully state the conditions and proofs for Theorem 3.3.

6)  In abstract and introduction, authors have defined federated learning as a setting over mobile devices. Although mobile devices takes a big part in federated learning, it is not the whole picture at all. Federated learning could be among non-mobile clients such as hospitals. Some minor wording might be needed to more accurately reflect the story of federated learning.

---

### Decision · Action_Editor_mowQ · 2024-09-08

**Recommendation:** Reject

**Comment:**

The paper proposed an approach to federated learning, where each edge device may have varying memory and network width which is a more realistic setting in practice.  While the reviewers agree that the problem is important to address, the reviewers have raised various points for improving the manuscript.  Some points were addressed in the revision phase. Nevertheless, in the final recommendation stage, reviewer sKQA has reserved opinions about the paper with concerns on the lacking insights and ablation study in the experiments s and the quality of the analysis. Reviewer PMrb  still expressed concerns about the validity of Assumptions C.1, C.2. Both reviewers mentioned above found the proof is not self-contained and is not easily accessible.   Reviewer 6BnQ voted for a weak acceptance.

I concurred with reviewers sKQA and PMrb that the paper needs a further substantial revision.  Therefore, I can not recommend its acceptance.

**Audience:**

The paper studies the federated learning -- a topic will certainly attract interests from the machine learning community.

**Claims And Evidence:**

As Reviewer sKQA commented in the final recommendation, the claims for the empirical improvement remain unclear and a comprehensive ablation study is not provided in the revised version.

**Resubmission Of Major Revision:**

The authors may consider submitting a major revision at a later time.